# Multiplexed detection of viral antigen and RNA using nanopore sensing and encoded molecular probes

Ren Ren [1,2,13], Shenglin Cai [1,3,13] ✉, Xiaona Fang[4], Xiaoyi Wang [1], Zheng Zhang[4], Micol Damiani[1], Charlotte Hudlerova [1], Annachiara Rosa [5,6], Joshua Hope [5], Nicola J. Cook [5], Peter Gorelkin[7], Alexander Erofeev[7], Pavel Novak[8], Anjna Badhan[9], Michael Crone [10], Paul Freemont [10], Graham P. Taylor [9], Longhua Tang [11], Christopher Edwards[2,8], Andrew Shevchuk[2], Peter Cherepanov [5,9], Zhaofeng Luo[4], Weihong Tan [4] ✉, Yuri Korchev [2,12], Aleksandar P. Ivanov [1] ✉ & Joshua B. Edel [1] ✉

We report on single-molecule nanopore sensing combined with position-encoded DNA molecular probes, with chemistry tuned to simultaneously identify various antigen proteins and multiple RNA gene fragments of SARS-CoV-2 with high sensitivity and selectivity. We show that this sensing strategy can directly detect spike (S) and nucleocapsid (N) proteins in unprocessed human saliva. Moreover, our approach enables the identification of RNA fragments from patient samples using nasal/throat swabs, enabling the identification of critical mutations such as D614G, G446S, or Y144del among viral variants. In particular, it can detect and discriminate between SARS-CoV-2 lineages of wild-type B.1.1.7 (Alpha), B.1.617.2 (Delta), and B.1.1.539 (Omicron) within a single measurement without the need for nucleic acid sequencing. The sensing strategy of the molecular probes is easily adaptable to other viral targets and diseases and can be expanded depending on the application required.

Coronavirus disease 2019 (COVID-19) is an infectious disease caused by severe acute respiratory syndrome coronavirus 2 (SARS-CoV-2) that has swept the globe since first being identified in December 2019. As of November 2023, there were more than 770 million confirmed cases, and nearly 7 million deaths have been reported worldwide by the World Health Organization (WHO)[1]. Although mass vaccination programmes have been globally implemented to protect the population from severe illness and hospitalisation, immunity can only protect

[1]Department of Chemistry, Imperial College London, Molecular Sciences Research Hub, White City Campus, 82 Wood Lane, London W12 0BZ, UK. [2]Department of Metabolism, Digestion and Reproduction, Imperial College London, Hammersmith Campus, Du Cane Road, London W12 0NN, UK. [3]Yusuf Hamied Department of Chemistry, University of Cambridge, Lensfield Road, Cambridge CB2 1EW, UK. [4]The Key Laboratory of Zhejiang Province for Aptamers and Theranostics, Aptamer Selection Center, Hangzhou Institute of Medicine (HIM), Chinese Academy of Sciences, 310022 Hangzhou, Zhejiang, China. [5]The Chromatin Structure and Mobile DNA Laboratory, The Francis Crick Institute, London, UK. [6]Wolfson Education Centre, Faculty of Medicine, Imperial College London, London, UK. [7]National University of Science and Technology "MISIS", Leninskiy Prospect 4, 119991 Moscow, Russian Federation. [8]ICAPPIC Limited, The Fisheries, Mentmore Terrace, London E8 3PN, UK. [9]Molecular Diagnostic Unit, Section of Virology, Department of Infectious Disease, Faculty of Medicine, Imperial College London, London, UK. [10]Section of Structural and Synthetic Biology, Department of Infectious Disease, Faculty of Medicine, Imperial College London, London, UK. [11]State Key Laboratory of Modern Optical Instrumentation, College of Optical Science and Engineering, International Research Center for Advanced Photonics, Zhejiang University, 310027 Hangzhou, China. [12]Nano Life Science Institute (WPI-NanoLSI), Kanazawa University, Kakuma-machi, Kanazawa 920-1192, Japan. [13]These authors contributed equally: Ren Ren, Shenglin Cai. ✉e-mail: shenglin.cai15@imperial.ac.uk; tan@hnu.edu.cn; alex.ivanov@imperial.ac.uk; joshua.edel@imperial.ac.uk

individuals over a limited period and commonly declines after 6 months[2]. Furthermore, the emergence of several variants has boosted the transmissibility[3] among the public while at the same time evading the immunity gained naturally or by vaccination[4,5]. This has highlighted the need to develop rapid and accurate diagnostic approaches that can be implemented for point-of-care (POC) testing of SARS-CoV-2 and relevant variants to minimise virus transmission.

Currently, the benchmark testing for SARS-CoV-2 relies on the detection of viral RNA and is based on the quantitative reverse-transcription polymerase chain reaction (RT-qPCR) due to its high sensitivity[6]. However, RT-qPCR-based testing is not available in situ, which translates into significant processing time between sample collection and obtaining the test result, logistical challenges due to sample transportation and increased testing costs, leading to significant economic loss. To overcome these limitations, efforts have been directed towards developing CRISPR-[7–10], isothermal-[11,12] and/or electrochemical-based[13,14] methods to simplify the detection of SARS-CoV-2 for POC use. Recent advances in viral antigen testing (i.e., spike protein)[15] and serological antibody assays (i.e., IgG and IgM)[16], typically based on the enzyme-linked immunosorbent assays (ELISA)[17,18] or lateral flow assays (LFA)[19], has the potential to address this problem to some extent; however, the sensitivity of these assays is lower in part due to the lack of target amplification unlike in RT-qPCR.

he detection of viral RNA, antigen proteins, or antibodies are effective in slowing down the spread of SARS-CoV-2; however, existing technologies are mainly focused on analysing one target at a time with only limited multiplexing capability. The single output of these approaches only provides partial information on infectious status resulting in higher rates of false positives/negatives[6,20]. Compared to single target detection, multiplexed sensing allows analysis of different targets from the same virus, which potentially improves the accuracy for assessment of disease severity[21,22], or, alternatively, enables scaling up the simultaneous detection of a range of pathogens which might induce similar symptoms, such as SARS-CoV, SARS-CoV-2, MERS-CoV, and influenza[23]. Several previously reported studies have addressed this challenge in part by using optical[24] and electrochemical[22] methods; however, their multiplexing capacity is largely restricted by spectral properties of the fluorophores or by the limited number of electrochemical detection channels, respectively. Sequencing-based approaches[23,25] have also been reported to increase the multiplexity, but they generally involve sophisticated analytical instrumentation and a long turnaround time.

Furthermore, current approaches generally lack the ability to identify viral variants, particularly those with only single-base mutations, unless using genomic sequencing.[26,27] The associated process is, however, rather time-consuming, costly and requires significant infrastructure, which can significantly delay the isolation of infected individuals and hamper quick response that can be taken by the local government and/or communities towards newly emerging variants of concern (VOCs).

Nanopore sensing, when combined with molecular probes, is a promising approach that offers a potential solution to the above limitations. This approach enables the detection of individual molecules by recording changes in current as they pass through the nanopore electrokinetically[28–32]. Importantly, the selectivity can be tuned using molecular probes, such as DNA/peptide-based carriers[33–37] and nanoparticles[38,39], or by modifying recognition units on the pore surface[40–42]. Several research groups, including our own, have successfully integrated custom-designed DNA molecular carriers with nanopore sensing to achieve selective and efficient protein sensing in solution[36,43–49]. However, the current multiplexing capability of this technology remains limited[50]. Furthermore, the typical detection range of nanomolar[51] to sub-picomolar[34,52–54] concentrations often exceeds the requirements of certain real-world applications, such as

the detection of SARS-CoV-2 in clinical settings, where concentrations can be as low as attomolar levels[22].

To address this drawback of enhanced nanopore sensing, we herein report an encoded DNA molecular probe for use with single-molecule nanopore sensing to simultaneously detect spike (S) protein on the viral surface and nucleocapsid (N) protein in the viral core, as well as multiple viral RNA genomic fragments of SARS-CoV-2, including emerging variants. The probe consists of a double-stranded DNA carrier with aptamers or complementary DNA oligos attached at specific positions, creating binding sites specific to each target. These probe molecules are introduced directly into a lysed or PCR-amplified clinical sample, where they can bind to the targets and are subsequently read using nanopore sensing. Upon binding, the presence (or absence) of viral S and N proteins and genomic RNA fragments can be quantified by detecting secondary peaks superimposed on the signal originating from the molecular probe alone, as shown in Fig. 1. The fractional position of these secondary peaks enables demultiplexing of the signal and accurate identification of the targets. Using this strategy, we can quantify the presence and relative abundance of S and N proteins directly from human saliva and nasal/throat swabs or RNA amplicons from N, S, and open-reading frame (ORF1b) genes of SARS-CoV-2. Although no direct nanopore sequencing is used, we show that the approach can discriminate variants with a single point mutation, and we demonstrate the possibility of discriminating viral variants: wild-type, Omicron (B.1.1.529), Delta (B.1.617.2), Alpha (B.1.1.7) in a single test without the need for genomic sequencing. We further validate the efficiency of the method by using pre-clinical and clinical samples, as confirmatory of the ability to differentiate among patients with SARS-CoV-2, pseudovirus controls and healthy controls.

## Results and discussion
### Nanopore sensing of the spike protein
Molecular probes consisting of 10 kbp DNA along with an S protein binding aptamer (SBA: 5′-CACGCATAACGTCTTGCGGGGCGGCGGG TTGAGAGGATGTCGGGTGGTTATGCGTG-3′) were used to target the S protein of SARS-CoV-2, Fig. 2a. The SBA was obtained synthetically using systematic evolution of ligands by exponential enrichment (SELEX)[55]. The SBA had a relatively high affinity towards the S1 unit of SARS-CoV-2 spike protein, responsible for recognition of and binding to the host receptor angiotensin-converting enzyme 2, as indicated by a low sub-nM dissociation constant ($K_D$), see Supplementary Fig. S1. The SBA was integrated into a customised 10 kbp dsDNA, obtained by enzymatically digesting 48.5 kbp lambda-phage DNA (λ-DNA) through base-pairing and ligation. Preparation protocols and gel characterisation are given in the "Methods" and Supplementary Fig. S2. Using this molecular probe, we next performed nanopore experiments to detect the binding between the S protein and the SBA-modified molecular probes. To optimise the signal-to-noise ratio of nanopore measurements, experiments were performed in a 2M LiCl buffer solution. Nanopores were fabricated by laser-assisted pulling of quartz capillaries[56]. The average diameter of nanopores used in this work is estimated to be $15 \pm 3$ nm as characterised by scanning electron microscope (SEM) images (Supplementary Fig. S3), which is in good agreement with the size calculated from conductance measurements, $15.4 \pm 1.2$ nS in 2M LiCl ($n = 20$), Supplementary Fig. S3.

Figure 2a illustrates a comparison of single-molecule ionic current signals obtained from the translocation of molecular probes (200 pM), both with and without the bound S protein (S1 region). In the absence of S protein, typical events with a single current level are observed. However, when S protein is present at a concentration of 20 nM, a distinct secondary current peak appears at either the beginning or end of the event, depending on the orientation of the probe during nanopore translocation. These secondary peaks exhibit an average amplitude of $85.2 \pm 23.8$ pA and a dwell time of $25 \pm 18$ µs. It is worth mentioning that a fraction of the observed signal events correspond to

partial folding, both in the absence and presence of S protein (Supplementary Fig. S4). However, these secondary peaks resulting from probe folding can be easily distinguished by their significantly smaller blockade current of 32.7 ± 9.9 pA, as shown in Fig. 2c and Supplementary Figs. S5 and S6. This smaller peak current can be attributed to the narrower width of the folded DNA double-helix compared to the size of the S protein (see Supplementary Fig. S7a for the dimensions of the S protein). Control experiments were conducted using S protein alone, without molecular probes, resulting in minimal translocation events (Supplementary Fig. S7b). To avoid counting false positives associated with folded events (63.2 ± 1.9%), we isolated S protein events by setting the following thresholds: (1) fractional peak position <0.1 ± 0.1 or >0.9 ± 0.1, to take into account the S protein bound to the DNA either being translocated from the tail or head ends. (2) Secondary peak width <0.2 ms and height larger than 1.5× the amplitude of the DNA level (Supplementary Fig. S8). The secondary peak current amplitude of folded events was consistently between 0.6× and 0.8×

the amplitude of the unfolded DNA level. It should be noted that when the translocation event exhibited both folding and protein binding signals, these events were individually examined to verify that all binding events were classified correctly.

Single-molecule nanopore measurements allow extraction of the binding ratio (%) from the fraction of bound to total events, even at concentrations lower than the $K_D$. The binding assay was performed to quantify the binding ratio (%) within the protein concentration range of 0–200 nM (Fig. 2b, d and Supplementary Fig. S9). The binding ratio for S protein increased initially and plateaued after 20 nM. A calibration curve was obtained, with the limit of detection (LOD) and limit of quantification (LOQ) being estimated to be 0.2 and 0.38 pM based on three and ten times the standard deviations above the background noise (Fig. 2d). The sensitivity could be further improved by increasing the capture frequency, for instance, by introducing an asymmetric salt concentration across the nanopore[57] or using nanopore detection in combination with dielectrophoresis[58]. The specificity of the SBA was

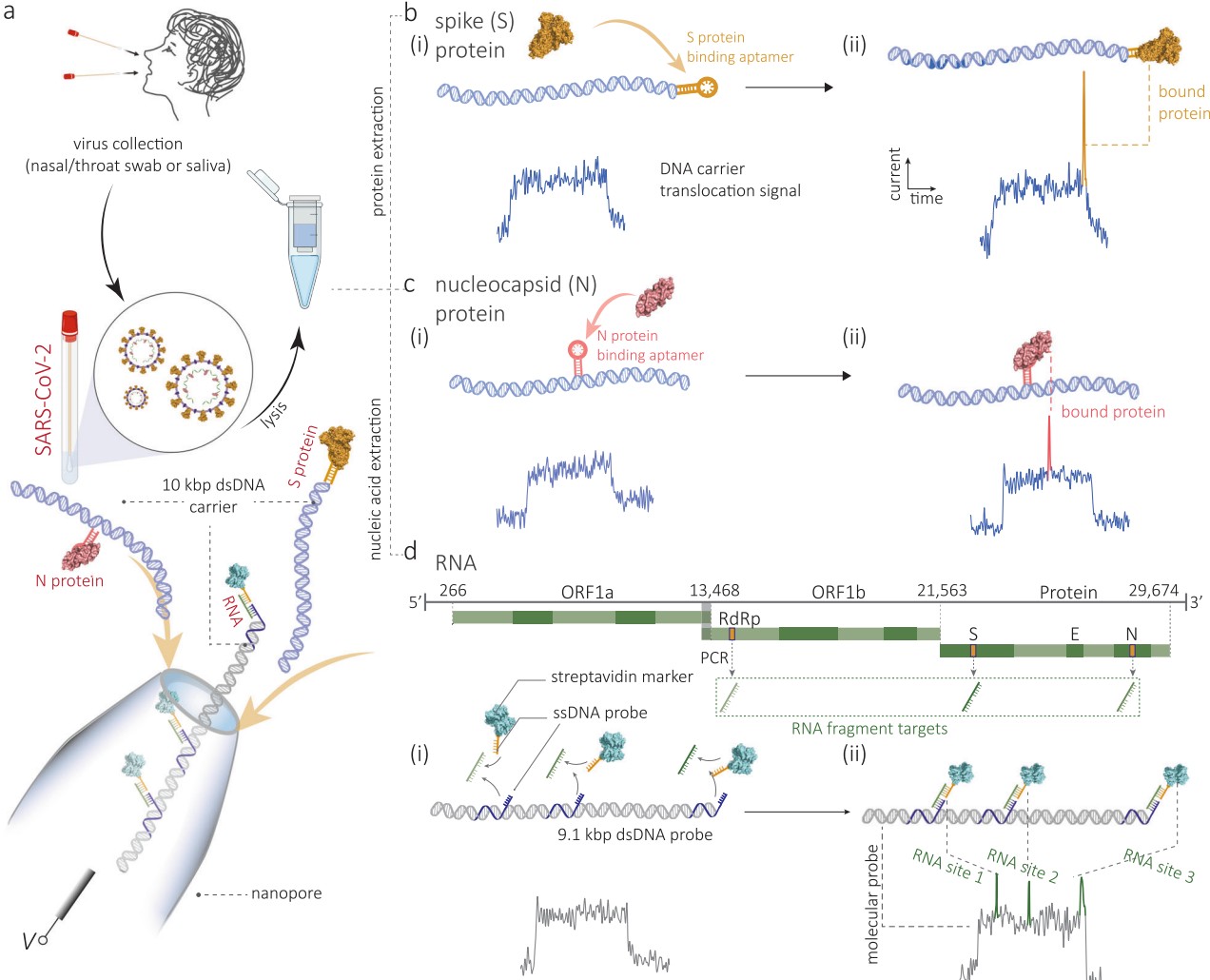

**Fig. 1 | Multiplexed sensing of SARS-CoV-2 S, N proteins and RNA fragments. a** Schematic of multiplexed detection of viral proteins and RNAs from patients' samples. SARS-CoV-2 virus is lysed into S protein, N protein, and RNA fragments, which can be selectively determined by the nanopore when bound to their corresponding encoded DNA molecular probes. **b** Schematic and representative ion current-time traces for the translocation of 10 kbp dsDNA probes encoded with S protein binding aptamer (SBA), without (**i**) and with (**ii**) bound S protein (**b**). **c** Schematic and representative single-molecule ion current-time signatures for the translocation of 10 kbp dsDNA probes encoded with N protein binding aptamer

(NBA) without (**i**) and with (**ii**) bound N protein (**c**). Bound S protein induces a distinguishable secondary peak at the end (**b**) while the N protein binding results in a secondary peak in the middle (**c**). **d** Schematic representations of a DNA probe (9.1 kbp) that has been encoded with 3 ssDNA sequences complementary to chosen regions in the ORF1b, S, and N genes of viral RNA. Sequence-specific binding of viral RNA fragments is identified by the presence of secondary peaks and their position in the ion current single molecule signature. Peaks are further enhanced by adding streptavidin-tagged biotinylated-ssDNA probes for signal enhancement with sequences complementary to the chosen regions in ORF1b, S, and N genes.

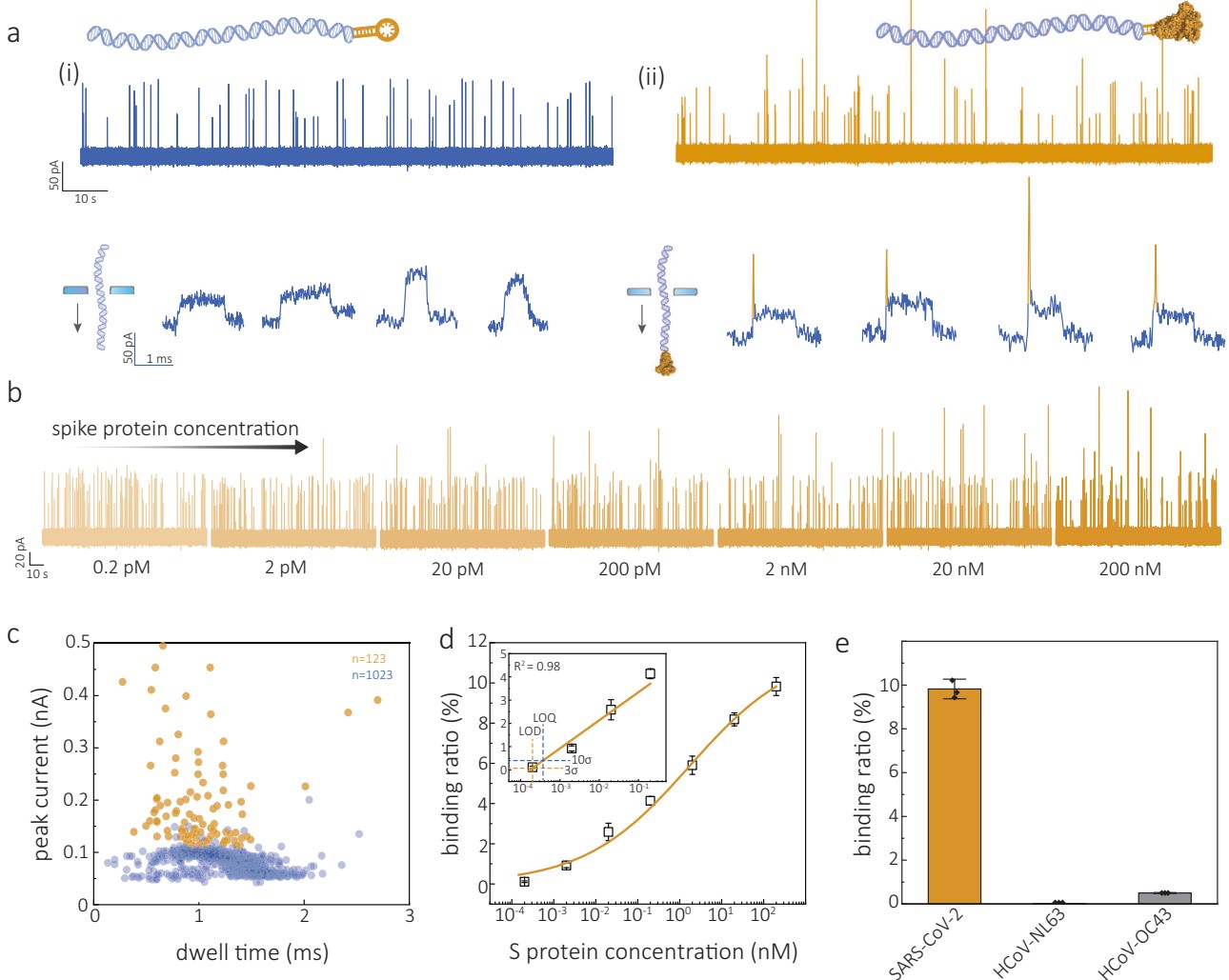

**Fig. 2 | Detection of S protein. a** Representative current-time traces for the translocation of SBA encoded 10 kbp DNA molecular probe (end-modified) in the absence (i) and presence (ii) of S protein (20 nM). Zoom-in views of typical translocation events are shown below current-time traces. **b** Representative current-time traces for concentration-dependent experiments with the addition of S protein of 0.2, 2, 20, 200 pM, 2, 20, and 200 nM, respectively. **c** Density scatter plots of dwell time versus peak current amplitude for the translocation of the molecular probe only (blue, $n = 1023$) and protein/DNA conjugates (orange, $n = 123$). **d** Binding curve of molecular probe incubated with S protein ranging from 0 to 200 nM. **e** Comparison of the detection of S protein (200 nM) from SARS-CoV-2 virus, HCoV-NL63 and HCoV-OC43. All translocation experiments were performed with 200 pM molecular probe in 2 M LiCl buffer (5 mM MgCl$_2$, 10 mM Tris–HCl, 1 mM EDTA, pH = 8) at an applied potential bias of 300 mV. Error bars in **d** and **e** represent the standard deviation of three independent replicates and the measure of the centre represents their corresponding mean value. Source data are provided as a Source Data file.

also assessed by performing measurements from other human coronavirus types. We observed a much lower binding ratio from HCoV-NL63 or HCoV-OC43 (0.02 ± 0.01% and 0.50 ± 0.03%) when compared with that obtained for SARS-CoV-2 (9.82 ± 0.45%), indicating excellent specificity of the selected aptamer and method, Fig. 2e and Supplementary Fig. S10.

**Nanopore sensing of the nucleocapsid protein**

We next investigated the sensing of N protein using an N protein binding aptamer (NBA: 5′-GCTGGATGTCGCTTACGACAA-TATTCCTTAGGGGCACCGCTACATTGACACATCCAGC-3′) encoded molecular probe (10 kbp). The NBA was obtained through a similar SELEX process similar to that of SBA with a $K_D$ of ≈0.5 nM as reported previously[59] (Supplementary Fig. S11). To perform simultaneous experiments and to distinguish the signal from the S protein, the aptamer was encoded in the middle of the molecular probe. Further details are provided in the "Methods" section and Supplementary Fig. S12.

The translocation signal of the molecular probes in the absence of N protein produces typical current blockades for 10 kbp DNA (Fig. 3a). Detailed statistics are summarised in Supplementary Figs. S13 and S14. In the presence of N protein, secondary peaks at the midpoint of the current blockades were observed owing to the formation of the protein–DNA complex, Fig. 3a and Supplementary Fig. S15. Much like with S protein, to avoid counting false positives, we isolated N protein events by setting the fractional peak position threshold to 0.5 ± 0.2 and secondary peak width <0.3 ms. Secondary peak amplitude was not used to discriminate between events due to the similarity in amplitude between DNA-folding and protein-bound events. The observation of DNA knots in the middle of the translocation event is uncommon (<0.1%), and hence all protein-bound events could be isolated based on the above two parameters. Similarly, all folding events were cross-checked manually to ensure all binding events were counted as partial folding will lead to a slight shift in the fractional position. The secondary peaks in these N protein-bound events exhibited a peak amplitude of 51.1 ± 13.9 pA and dwell time of 23 ± 12 μs.

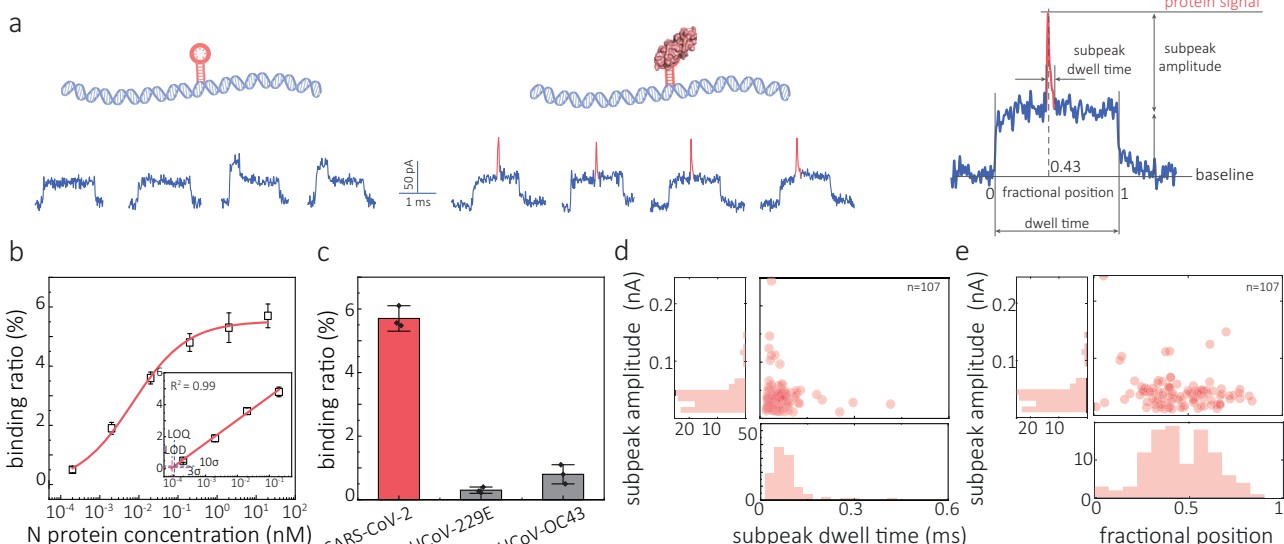

**Fig. 3 | Nanopore detection of N protein. a** Schematics of NBA modified 10 kbp DNA molecular probe and its binding to N protein. Representative translocation events are shown underneath. Upon binding, the N protein produces a secondary peak superimposed on the dsDNA level at a fractional position of ≈0.43 for a typical event. **b** Binding curve of molecular probe incubated with N protein ranging from 0 to 20 nM. **c** Comparison of the detection of N protein (20 nM) from SARS-CoV-2 virus and seasonal flu virus of HCoV-229E and HCoV-OC43. **d, e** Scatter plots of

secondary peak amplitude versus dwell time (**d**) or fractional position (**e**) and corresponding statistics (*n* = 107). All the translocation experiments were performed with 200 pM molecular probe in 2 M LiCl, 5 mM MgCl₂, 10 mM Tris–HCl, 1 mM EDTA, and pH 8 solution at an applied bias of 300 mV. Error bars in **b** and **c** represent the standard deviation of three independent experimental repeats and the measure of the centre represents their corresponding mean value. Source data are provided as a Source Data file.

Notably, this amplitude is smaller than that of the S protein secondary peak (85.2 ± 23.8 pA), which can be attributed to the smaller size of the N protein, Fig. 3d and Supplementary Fig. S13. To estimate the relative position of protein binding, we quantified the fractional position of the secondary peak with '0' being defined as the beginning and '1' as the end of the translocation event, respectively, Fig. 3a. The average fractional position for the bound N protein was calculated to be 0.48 ± 0.21 (Fig. 3e), which is in good agreement with the position of the NBA in the dsDNA probe (4931 in 10,201 bp), Supplementary Fig. S12. The LOD and LOQ were determined to be 0.09 pM and 0.11 pM, Fig. 3b and Supplementary Fig. S16. We also examined the NBA specificity by comparing two seasonal coronaviruses, HCoV-229E and HCoV-OC43. The binding ratio was observed to be much lower for HCoV-229E and HCoV-OC43 N protein (0.31 ± 0.11% and 0.83 ± 0.32%) as compared to that of SARS-CoV-2 (5.71 ± 0.42%) (Fig. 3c and Supplementary Fig. S17), revealing an excellent specificity towards SARS-CoV-2.

To verify the ability of multiplexed detection, we conducted nanopore translocation experiments by incubating S protein (20 nM) and N protein (20 nM) with a complex of SBA and NBA molecular probes (200 pM each). The results showed that S protein-positive events exhibited a secondary peak at the end of the translocation events, while N protein-positive events displayed a secondary peak in the middle, Supplementary Fig. S8.

**Multiplexed detection of S, N, and ORF1b genes of SARS-CoV-2**
Taking advantage of the sub-molecular resolution offered by nanopore sensing, we developed a detection strategy for simultaneously detecting the S, N, and ORF1b genes. This approach involved the use of a single DNA molecular probe that was encoded with short DNA oligos complementary to specific regions of each target gene fragment, as shown in Fig. 4. We designed three probes that can target RNA fragments transcribed from ORF1b, S, and N genes and integrated them into a 9.1 kbp dsDNA molecular probe at fractional positions of 0.18, 0.48 and 1, respectively, along the dsDNA probe length, Fig. 4a, b. Detailed information regarding the design, modification, and

preparation of this molecular probe can be found in the "Methods" section and Supplementary Figs. S18 and S19. In this strategy, streptavidin-tagged biotinylated single-stranded DNA (ssDNA) probes were also introduced. These probes could form a bridge with the target gene fragment, enhancing the ion current signal readout when they were transported through the nanopore, Fig. 4b(i). It is important to note that this strategy aims to detect RNA amplicons transcribed from specific genomic regions rather than targeting the entire viral RNA. This is due to the complex secondary structure of long RNA (~30 kb in this case), which makes it challenging for the probes to access and bind to the region of interest[60]. Additionally, in real-world settings, unprocessed RNA is often present in relatively low concentrations[61]. The amplification step helps increase the overall abundance of the target fragments, leading to improved sensitivity in the detection process. Given that PCR is a well-established technique, we combined it with nanopore detection in this study to demonstrate the proof-of-concept. However, the method is compatible with simpler approaches like loop-mediated isothermal amplification (LAMP) and rolling circle amplification (RCA), which could serve as potential alternatives. The main benefit of combining these techniques with nanopore-based detection is the enhanced multiplexing capabilities that allow the combined detection of multiple proteins and nucleic acids from the same sample. This permits the simultaneous detection of multiple targets and the differentiation of key mutations within a single test, thereby removing the necessity for sequencing.

To verify this detection method, we initially performed translocation experiments using synthetic RNA targets (ORF1b, S, and N site) at a concentration of 2 nM each and incubated the sample with molecular probes at a concentration of 200 pM, Fig. 4b. In the presence of the ORF1b, S, or N gene, individually, we observed the respective secondary peaks occurring in the corresponding fractional positions of 0.21 ± 0.07, 0.55 ± 0.11, and 0.98 ± 0.02, respectively. These values are in good agreement with the expected positions of 0.18, 0.48, and 1 for the ORF1b, S, and N sites, respectively. Control experiments were conducted using the molecular probe alone (200 pM) and the molecular probe (200 pM) in combination with

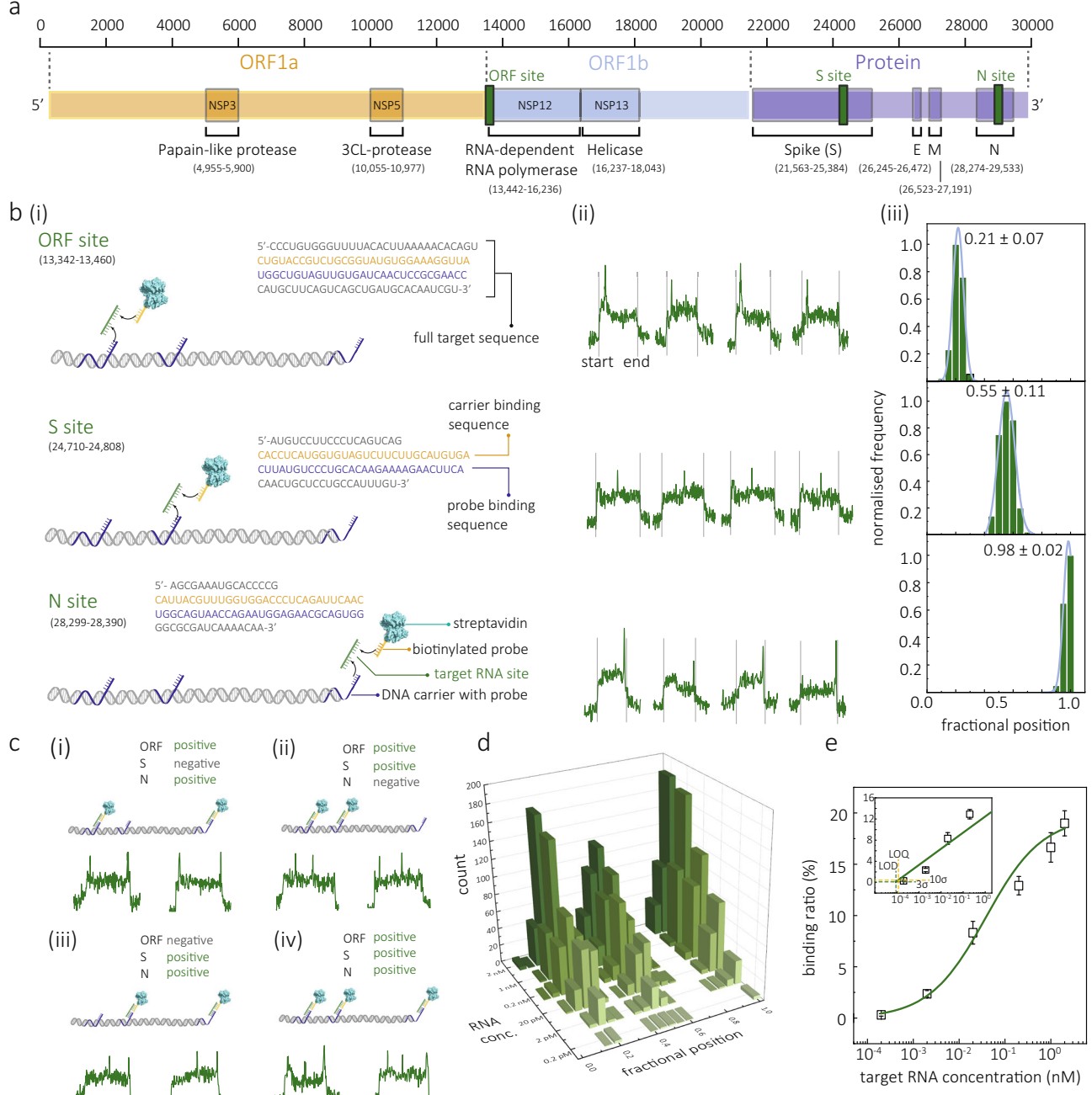

**Fig. 4 | Multiplexed detection of SARS-CoV-2 RNA fragments. a** Genome map showing the full length with annotated regions of SARS-CoV-2. Position of sequences in the ORF1b, S and N genes that have been chosen for detection are marked in magenta. **b** (i) Schematic representations of a 3-site DNA molecular probe (9.1 kbp) and the binding to ORF1b, S, and N gene targets, respectively. The probe for an individual target is assigned to a specific position along the dsDNA and bound to the first half of the target RNA sequence. A biotinylated sequence is used as a signal enhancement probe to bind the second half of the target sequence, and streptavidin is used to enhance the signal. Sequences for the selected ORF1b, S, and N gene targets are shown, and the binding segment of the dsDNA carrier and biotinylated probe are highlighted in orange and blue, respectively. (ii) Representative events are shown for the individual genome target binding with secondary peaks and are observed at the respective position. (iii) histogram of the

normalised frequency of the fractional secondary peak position for each binding genome fragment (*n* = 100). The detailed secondary peak information is shown in Supplementary Fig. S21. **c** Schematics for the 3-site molecular probe binding to two (i-iii) or three (iv) target gene fragments and the resulting translocation events are shown in the bottom panel. **d** Statistics for simultaneous detection of the three RNA targets range from 0.2 pM to 2 nM. **e** Calibration curve for the N gene binding ratio and the target RNA concentration. All the translocation experiments were performed with 200 pM of molecular probe in 2 M LiCl buffer (5 mM MgCl₂, 10 mM Tris–HCl, 1 mM EDTA, pH = 8) at an applied potential bias of 300 mV. Error bars in **e** represent the standard deviation of three independent experimental repeats and the measure of the centre represents their corresponding mean value. Source data are provided as a Source Data file.

streptavidin-tagged secondary probes (4 nM). In both cases, only translocations without secondary peaks were detected, indicating a minimal occurrence of false positives (0.1 ± 0.1%), Supplementary Fig. S20. Furthermore, we exposed the molecular probe to either two

or all three target gene sites, as shown in Fig. 4c. The resulting signals exhibited secondary peaks that aligned with their respective positions.

Subsequently, we assessed the sensitivity by conducting a titration curve and quantifying the binding ratio across a range of synthetic

RNA concentrations, spanning from 0.2 pM to 2 nM, as depicted in Fig. 4d. We obtained a calibration curve spanning five orders of magnitude in concentration, as illustrated in Fig. 4e and Supplementary Fig. S22. The corresponding LOD and LOQ were estimated to be 0.09 and 0.11 pM, respectively. Experiments were also performed with full-length synthetic SARS-CoV-2 RNA (wild-type strain, Twist Bioscience, USA). For all three genes, 35-cycle PCR amplification was performed; see the "Methods" section and for the primer design and gel characterisation. The lowest detected concentration was 0.01 copies/μl (Supplementary Figs. S23 and S24), which represents approximately two orders of magnitude lower than the more commonly used gold standard, RT-qPCR[7,62]. It should be noted that the calibration curve was constructed only for the N gene, while the other two binding sites were used to verify the presence of the other two genes, not for their quantification.

### Discrimination of SARS-CoV-2 variants of concern

We also examined the potential specificity of this sensing strategy for the discrimination of mutation points associated with various emerging variants. We initially chose the mutation D614G, as the target. The D614G is a vital mutant of which the 614th amino acid in the spike protein mutates from aspartic acid to glycine, resulting from a single-nucleotide A > G mutation at position 23403 of the genome[63,64]. This mutation has been reported to increase the infectivity and stability of virions, resulting in enhanced viral replication in human cells and tissues associated with higher viral loads in the upper respiratory tract of COVID-19 patients in nasal washes and trachea, and has the potential to increase transmission[64,65]. The D614G mutation was present in multiple VOCs, such as B.1.1.7, B.1.617.2, and B.1.1.529 (or Alpha, Delta, and Omicron variants, respectively)[66]. To identify this mutant, a primary probe able to distinguish the D614G gene fragment was designed and attached in the middle of a 9.1 kbp molecular probe (Fig. 5a,b). We tested this molecular probe with amplified D614G (from full-length Delta variant) gene and wild-type gene, respectively. The primers for the D614G site can be found in Supplementary Table 5. To achieve an optimised probe to distinguish the single-nucleotide mutation, we tested various lengths of the primary probes ranging from 9-nucleotide (nt) to 21-nt (see sequences in Supplementary Table 5). The results show that the 13-nt probe (mismatched point in the centre) had the highest potential to differentiate between the mutant and wild-type genes, Supplementary Fig. S25.

Based on this concept, we sought to design a scheme to discriminate SARS-CoV-2 variants by encoding probes that could identify characteristic mutations. For example, in addition to D614G, Y144del and G446S are also important mutations and are associated with the Alpha[67] and Omicron[68] variants, respectively (Fig. 5c). Therefore, we designed a 3-site molecular probe (probe 1) with sites 1, 2, and 3 labelled to target Y114del, D614G, and N gene (a highly conserved gene among SARS-CoV-2 variants), respectively (Fig. 5d). Another molecular probe (probe 2) was also designed by replacing the Y144del probe in site 1 with a probe targeting G446S. All probe designs and sequences are shown in Supplementary Tables 5–7. We tested DNA probe 1 with amplicons from the wild-type, Alpha, and Delta variants of SARS-CoV-2, and probe 2 with amplicons from wild-type, Delta, and Omicron variants (Fig. 5e). We observed that all variants, including wild-type, were positive for the N gene at the fractional position of 1, with the binding ratio of $15.1 \pm 1.9\%$, $13.2 \pm 2.1\%$, $14.6 \pm 1.8\%$ and $12.9 \pm 3.1\%$ for wild-type, Alpha, Delta, and Omicron variants, respectively, Fig. 5e and Supplementary Fig. S26–S29. Further, all three variants (Alpha, Delta, Omicron) showed positive hits for the D614G mutation at a fractional position of ca. 0.5 with binding ratios of $12.3 \pm 2.3\%$, $12.6 \pm 3.1\%$ and $11.8 \pm 3.3\%$, respectively, Fig. 5e(ii–iv). However, only negligible binding events ($0.1 \pm 0.1\%$) can be observed for the wild-type strain. In addition, the Alpha variant also showed positive hits for Y144del ($13.6 \pm 2.1\%$), and the Omicron variant ($11.1 \pm 1.1\%$) showed positive hits for G446S,

while almost no events (across 2000 events) were recorded for wild-type ($0.1 \pm 0.0\%$) and Delta ($0.1 \pm 0.1\%$) strains. Collectively, these results demonstrate excellent selectivity in identifying key mutations and exhibit the potential to rapidly discriminate among SARS-CoV-2 variants without the need to perform RNA sequencing.

### Pre-clinical test of S and N protein in saliva, S protein from pseudovirus, and full-length RNA of SARS-CoV-2

We have previously demonstrated the advantage of detecting biomarkers directly in unprocessed biological matrices such as serum and urine using nanopore sensing in conjunction with molecular probes[33]. Here we demonstrate the potential of screening multiple viral antigen proteins from human fluids, such as saliva. Saliva is reported to be an abundant source of respiratory viral content, such as SARS-CoV-2[69], and a readily available medium for non-invasive liquid biopsies[70].

To verify the feasibility of our approach, we conducted translocation experiments using SBA and NBA molecular probes (200 pM each) in a pooled human saliva sample obtained from healthy individuals (more than three individuals). The sample was diluted in 2 M LiCl buffer at a ratio of 1:20. Supplementary Fig. S30 shows that in this biofluid, ion-current signatures of molecular probes are not affected by the presence of the complex media and that events are observed to be identical to typical translocation without the presence of secondary peaks. It should be noted that a small fraction of events (up to ≈8.8% at a 1:20 dilution ratio) were observed to have relatively short dwell times ($0.36 \pm 0.18$ ms) (Supplementary Fig. S31). These events can be attributed to the translocation of other background molecular species present in human saliva and were excluded from our analysis.

A colour-coded pixel grid is shown in Fig. 6a, where the first 240 molecular probe events are classified based on the presence of secondary peaks in their ion-current signature. When the S and N protein was spiked in the saliva (with a final concentration of 20 nM), translocation events associated with both the S protein (orange, secondary peak at either end) and the N protein (red, secondary peak in the middle) could be observed. For any subsequent analysis, a total of 2000 events were used, Supplementary Fig. S31.

After confirming the feasibility of our approach, we proceeded to validate the detection using a pseudovirus and full-length RNA of SARS-CoV-2 (Supplementary Fig. S32). Pseudovirus is a synthetic chimaera of viral analogue, consisting of a surrogate viral core derived from the parent virus, an envelope glycoprotein on the surface derived from a heterologous virus, and a modified genome with essential genes required for replication being deleted[71]. These pseudoviruses serve as useful models for studying their parent viruses, as they exhibit high similarity and relatively low toxicity, allowing for research in Biosafety Level 2 laboratories[71]. In this study, we utilised the pseudovirus of SARS-CoV-2 as the target for detecting antigen proteins. The pseudovirus (≈$10^6$ copies/μl) was first inactivated and lysed using a lysis and extraction buffer (with protease inhibitor) to release the S protein on the particles, see the "Methods" section for details. The resultant solution was then incubated with the prepared SBA molecular probe (200 pM) within 2 M LiCl buffer at a ratio of 1:50, and nanopore experiments were then performed (Supplementary Fig. S32a). As shown in Supplementary Fig. S32a–d, probe signatures with secondary peaks were observed (0.6% over a total of 5000 detected probes), revealing the presence of S protein in the solution. According to the calibration curve in Fig. 2d, we estimated the concentration of S protein to be ~0.5 pM (Supplementary Fig. S32). The calculated number of S proteins per virus was ca. 31.3, which agrees with previously reported values in similar conditions (ca. 31–35)[72].

Gene sequence analysis using full-length synthetic RNA (Twist Bioscience, USA) was also assessed. Prior to testing, the RNA genome (wild-type, 10 copies/μl in nuclease-free water) was first amplified using a standard PCR process (35 cycles) with designed primers for the target ORF1b, S, and N gene sites and then downstream of transcription using

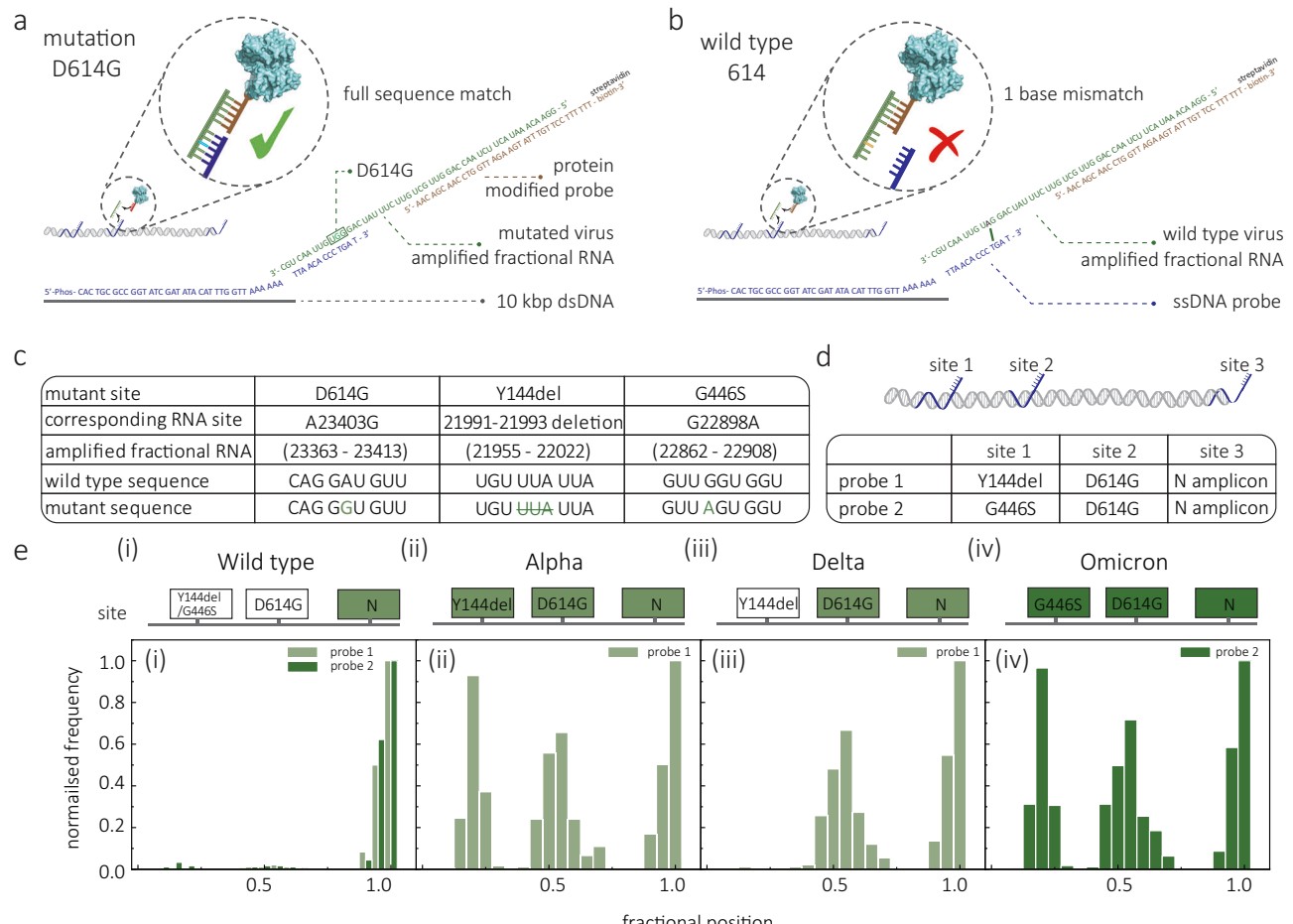

**Fig. 5 | Rapid differentiation of SARS-CoV-2 variants. a, b** Schematic illustration showing the strategy used to identify the single-nucleotide mutant of D614G mutation (**a**) and SARS-CoV-2 virus wild-type (**b**). The molecular probe and the signal enhancement probe are encoded to bind the D614G mutation sequence but not the corresponding sequence fragment in the wild-type. **c** Sequence information for three vital mutations of SARS-CoV-2: D614G, Y144 deletion, and G446S. **d** Design of two molecular probes encoded for mutations of Y114del/G446S, D614G, and N gene at sites 1, 2, and 3. **e** List of the mutation sites for the wild-type, Alpha, Delta, and Omicron variants as well as the corresponding mutations they contain. Fractional secondary peak positions were recorded for wild-type, Alpha, Delta, and Omicron variants. Molecular probe 1 was tested for wild-type, Alpha, and Delta variants, and molecular probe 2 was tested for wild-type, Delta, and Omicron variants. All the translocation experiments were performed with 200 pM of molecular probe in 2 M LiCl buffer (5 mM MgCl$_2$, 10 mM Tris–HCl, 1 mM EDTA, pH = 8) at an applied bias of 300 mV. Source data are provided as a Source Data file.

T7 transcriptase to obtain single-stranded RNA, Fig. 6b(i), see the "Methods" section for details. After incubation with the 3-site molecular probe, nanopore measurements showed events with corresponding secondary peaks in all three positions (≈18.4% over 5000 events), indicating the presence of target ORF1b, S, and N genes in the sample, Fig. 6 (ii-iv). We estimated the concentration of the RNA to be 1.96 ± 0.34 nM based on the calibration curve in Fig. 4e. The detection solution contained 1% of lysis lysate (with RIPA buffer and protease inhibitor) for the pseudovirus detection and 2% of PCR products (with PCR master mix buffer, reverse transcription buffer, etc.).

**Clinical test of patients with variants of concern**
We conducted tests on nasal/throat swab samples obtained from healthy controls as well as patients infected with the wild-type, Delta, and Omicron variants of the virus. Each sample was divided into two parts for protein and RNA analysis (Supplementary Fig. S33).

To detect proteins, the samples were initially lysed using RIPA lysis buffer to release the proteins. Subsequently, a purification step was performed using an Amicon Ultra Centrifugal filter with a molecular weight cutoff of 10 kDa. Further details regarding this process can be found in the 'Methods' section. The resulting samples were then subjected to testing using SBA and NBA-modified molecular probes.

For the healthy control samples ($n = 5$), neither the S protein nor the N protein was detected (Fig. 7a(i) and Supplementary Figs. S34 and S35). However, when testing the patient samples infected with the wild-type ($n = 5$), Delta ($n = 5$), and Omicron ($n = 5$) variants, an increased binding ratio was observed at the fractional positions of 0.5 and 1.0 for all these samples (wild-type: P6–10, Delta: P11–15, and Omicron: P16–20). This finding indicates the presence of both S and N proteins in the respective patient samples.

For RNA detection, the nasal/throat samples were pyrolysed at 95 °C for 5 min to release the RNA, followed by 35-cycle PCR amplification for target gene regions containing G339D mutation, D614G mutation, and N gene. The resultant RNA amplicon was then incubated with a 3-site DNA molecular probe (9.1 kbp), which was encoded to target G339D, D614G, and N gene, respectively, and tested by running nanopore experiments (details and sequences of probes and primers provided in 'Methods') Similar to the protein testing, all the patient samples (P6-P20) that had previously tested positive using RT-qPCR were confirmed positive for the N gene, while only a few binding events were observed for the healthy controls, Fig. 7a(ii) and Supplementary Figs. S34 and S35. Further, by analysing the fractional position of secondary peaks for a typical sample from each group, we observed a positive hit for the N gene (fractional position of 1.0) in patients with

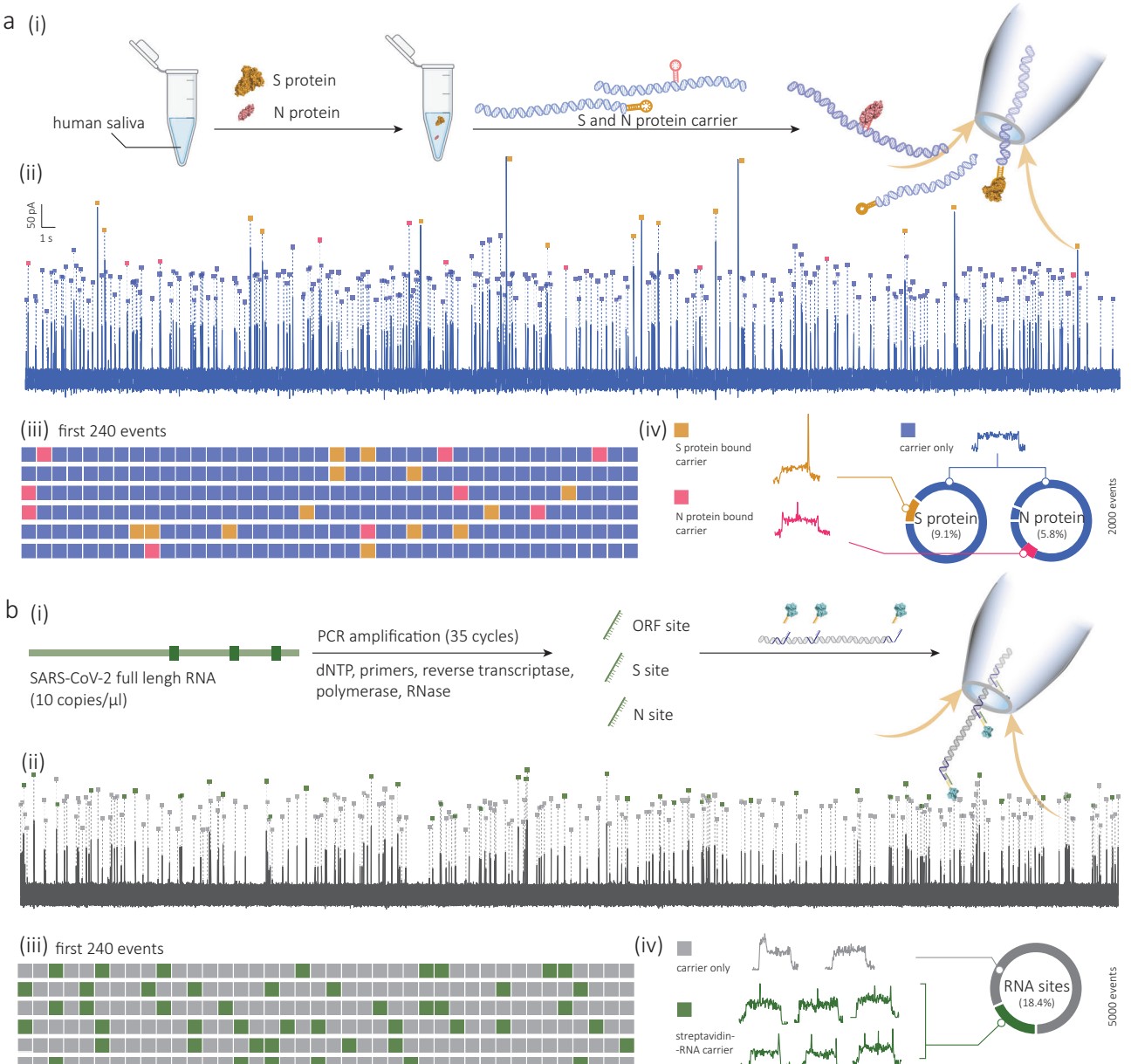

**Fig. 6 | Digitally multiplexed sensing of S and N proteins in saliva and full-length RNA of SARS-CoV-2. a** Sensing of S and N proteins in human saliva. (i) Schematic showing the workflow of multiplexed S and N protein detection in human saliva. 20 nM S and N proteins are added in pooled human saliva (>3 people, 1:20) solution and incubated with the prepared SBA and NBA molecular probes, and nanopore measurements are performed. A representative current-time trace for the detection is shown with the molecular probes marked with blue squares, N protein binding events marked with red squares, and S protein binding events marked with orange squares (ii). The classification of the first 240 detected probes is shown in a colour-coded pixel grid (iii), and the breakdown of a total of 2000 detected probes is shown in (iv). **b** Multiplexed sensing of viral genes from full-length synthetic RNA of SARS-CoV-2. (i) Schematic showing the workflow of

detecting viral RNA. Target gene sites (i.e., ORF, S, and N gene) are pre-amplified from the whole viral RNA through a 35-cycle PCR by adding corresponding primers and enzymes. The amplified genome fragments are then incubated with the prepared 3-site molecular probe and detected by nanopore sensing. A representative current-time trace for the detection is shown with the molecular probes marked with grey squares, and the bound target RNA events are marked with magenta squares (ii). The classification of the first 240 detected probes is shown in a colour-coded pixel grid (iii), and the classification of a total of 5,000 detected probes is shown in (iv). All the translocation experiments were performed with 200 pM molecular probes in 2 M LiCl buffer (5 mM MgCl$_2$, 10 mM Tris–HCl, 1 mM EDTA, pH = 8) at an applied bias of 300 mV. Error bars represent the standard deviation of three independent experimental repeats.

wild-type (11.6 ± 1.8%), Delta (7.9 ± 2.0%), and Omicron strains (6.7 ± 1.3%) as opposed to the healthy controls (0.4 ± 0.2%), Fig. 7b. Furthermore, an additional positive hit for the D614G gene at position of ≈0.5 can be observed for the patients of Delta (6.9 ± 1.0%) and Omicron (4.3 ± 0.7%). In addition, the Omicron patient sample also showed a positive hit for G339D mutation (a characteristic mutant for Omicron[73]) at fractional position of ≈0.18 with binding ratio of

(4.6 ± 0.9%), while only a negligible binding ratio was observed for the healthy control (0.1 ± 0.1%), wild-type (0.2 ± 0.2%), and Delta (0.2 ± 0.1%). The nanopore results were in good agreement with control RT-qPCR measurements, Supplementary Table 9. These findings indicate that nanopore sensing with encoded molecular probes has the potential to distinguish among SARS-CoV-2 variants with single-base mutations based on patient samples. However, it is important to note

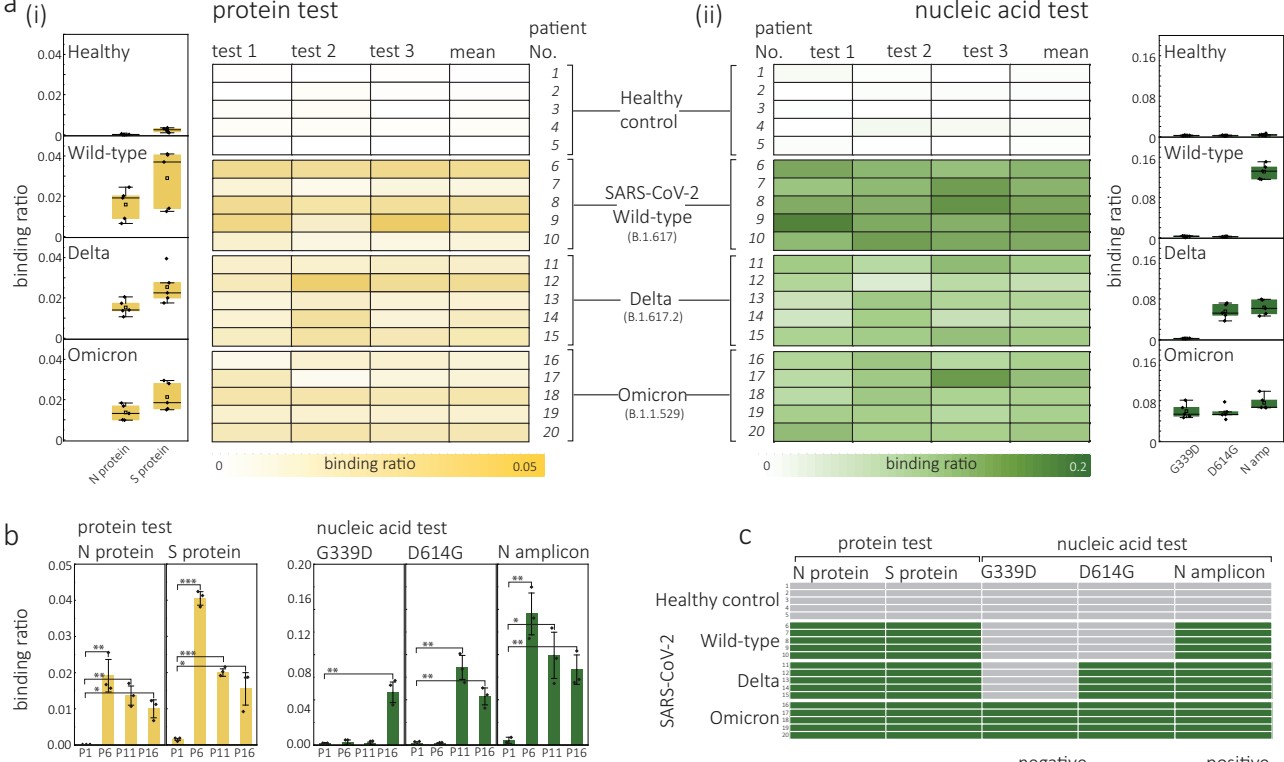

**Fig. 7 | Validation with clinical samples of different SARS-CoV-2 variants. a** Box plots and a colour map showing the binding ratio for the overall S and N protein (i) and the N gene (ii) from samples of healthy controls ($n = 5$), and patients infected by the wild-type ($n = 5$), Delta ($n = 5$), and Omicron ($n = 5$) variants. Box plots in (i) show the binding ratio for S and N proteins individually for each cohort of patients. Box plots in (ii) show the binding ratio for the N gene and mutation of D614G and G339D individually for each cohort of patients. Each patient sample was tested independently 3 times. For all boxes, the black central line represents the median, the square represents the mean, the bottom and top edges mark the 25th and 75th percentiles. Whiskers denote the intervals between the 5th and 95th percentiles.

**b**, Bar charts of the binding ratio of S/N protein (i) and G339D, D614G and N gene (ii) for one typical patient from each cohort. Statistical significance was tested using a two-tailed Student's $t$-test. *$P < 0.05$; **$P < 0.01$; ***$P < 0.001$, the detailed $P$-value information is shown in Supplementary Table 10. **c** Overall test results of S and N protein, G339D, G614G mutation, and N gene for each patient. All translocation experiments were performed with 200 pM molecular probes in 2 M LiCl buffer (5 mM $MgCl_2$, 10 mM Tris–HCl, 1 mM EDTA, pH = 8) at an applied bias of 300 mV. Error bars in **e** represent $1 \times \sigma$ obtained from three different nanopore measurements ($n = 3$). Source data are provided as a Source Data file.

that further clinical trials are required to validate this method before it can be applied in practical settings.

In summary, the combination of nanopore sensing and position-encoded DNA molecular probes allows for the simultaneous detection of various antigen proteins and multiple RNA gene fragments of SARS-CoV-2 with high sensitivity and selectivity. Our study demonstrated the ability of nanopore sensing to directly detect S and N proteins in unprocessed human saliva. Additionally, we achieved high sensitivity, detecting as low as 1 copy of RNA per 100 μl in patient samples obtained from nasal/throat swabs with a 35-cycle PCR amplification step. The strategy we employed enables the recognition of S or N proteins of SARS-CoV-2 across different coronavirus types. Moreover, it has the potential to identify key mutations, such as D614G, G446S, or Y144del, in viral variants with single-nucleotide polymorphism resolution. Notably, our approach allows for the detection and discrimination of different lineages of SARS-CoV-2, including the wild-type, B.1.1.7 (Alpha), B.1.617.2 (Delta), and B.1.1.539 (Omicron), in a single measurement without the need for nucleic acid sequencing.

We validated the detection of S protein using pseudovirus and the detection of N, S, and ORF genes. Furthermore, we successfully differentiated COVID-19 patient samples from healthy controls by simultaneously detecting antigen proteins and RNA genes, and we demonstrated the potential to discriminate among key mutations, such as D614G and G339D, associated with Delta and Omicron variants. It is important to note that further clinical trials and validation are

necessary before this method can be implemented in practical clinical settings.

Our strategy offers several potential applications. Firstly, since probes designed to detect mutations can be easily customised, this method can be quickly adapted and deployed when new sequenced variants emerge. This enables timely response by local authorities and provides accessible technology for surveillance testing of new variants. Secondly, this approach can be used for monitoring infection in individuals by simultaneously detecting a range of targets, including antigens, genomic RNA, and antibodies. Lastly, apart from SARS-CoV-2, this method can be readily expanded to detect other pathogens, such as influenza variants, which would be especially useful during the cold seasons and flu epidemics.

## Methods

### Ethical statement

The viral samples were provided from the Imperial College London testing scheme and were from fully anonymised, redundant samples (i.e., samples left over after testing) and retained for assay development, quality assurance, and validation. The sequencing was part of the Imperial College London's response to ensure that new variants were detected and to detect otherwise unexplained clusters. The consent to providing the sample and to the testing of the sample was provided at test booking through an online process. The use of virus samples was in accordance with RCPath guidelines.

## Aptamer selection

The DNA library contains a 76nt length ssDNA, of which 36 bases were random nucleotides, and the remaining 40 were two 20-base primers. The SELEX was then performed using the magnetic beads, and a quantitative polymerase chain reaction (qPCR) was used to monitor the selection process. Briefly, the S and N protein was coupled on the surface of NHS-activated carboxylic acid dynabeads (Invitrogen) following a typical EDC/NHS protocol. 100 μL of 500 nM DNA library was then incubated with the S/N protein-conjugated magnetic beads in the binding buffer containing 2 mM $KH_2PO_4$, 8 mM $Na_2HPO_4$, 150 mM NaCl, 5 mM KCl, 1 mM $CaCl_2$, 0.5 mM $MgCl_2$, pH 7.4. The protein-coated beads were washed using the binding buffer to remove unbound and weakly bound sequences. The bound DNA strands were subsequently recovered and amplified by PCR for a total of five rounds of selection. The enriched DNA pool was then identified through high-throughput sequencing.

## SPR measurement of aptamer binding to protein

The binding affinity of the selected aptamer to the target protein was measured using surface plasmon resonance (SPR) experiments conducted on a BIAcore T200 biosensor system from Cytiva. Using the standard amine-coupling method, the protein was immobilised directly onto a CM5 sensor chip. The binding kinetics analysis was carried out using the multi-cycle mode, wherein aptamers were injected at different concentrations with a flow rate of 30 μL/min. Association and dissociation phases were allowed to proceed for 2-3 minutes each. During the experiments, a running buffer of DPBS was used, and a regeneration buffer consisting of 1 M NaCl with 5 mM NaOH was employed to regenerate the sensor chip surface. Sensorgrams were obtained by varying the aptamer concentration, and subsequently, the BIA evaluation software was utilised to analyse the data and calculate the equilibrium dissociation constant ($K_d$). The obtained data were fitted to the Langmuir model for a 1:1 binding.

## UV–Vis measurements

The concentration and purity of newly ordered DNA/RNA oligos and in-house prepared DNA molecular probes were estimated by measuring the UV-Vis absorbance using a Nanodrop (Thermo Scientific). In brief, 1 μl of DNA/RNA solution was loaded onto the pedestal of the Nanodrop and absorbance was measured from 220 to 350 nm. The concentrations of oligos were calculated using the absorbance value at 260 nm and the extinction coefficient provided by the supplier. We measured the absorbance at 260 nm to estimate the molecular probe concentrations and calculated using an extinction coefficient of 0.02 ml•μg-1•cm-1. The purity of the obtained probe was assessed by calculating the ratio of A260/280 and A266/230.

## Gel electrophoresis for characterisation

The preparation of NBA-labelled DNA molecular probe using 10 kbp plasmid (Supplementary Fig. 12c) was characterised by gel electrophoresis. Briefly, 10 μl of product from each reaction step (containing ~50 ng of DNA) was firstly mixed with 2 μl of 6 × purple loading dye (New England BioLabs, UK). 10 μl of the above DNA solution and 1 kbp extended DNA ladder were loaded into the well of a 0.7% agarose gel. The electrophoresis was performed in TBE buffer at a potential of 5 V/cm for 120 min. Then, the gel was stained with 25 ml of 1 × SYBR Gold solution and incubated for 30 min under darkness. The gel was visualised using a Gel-Bright™ LED Light Box (Biotium™) and imaged using a camera.

## Preparation of DNA molecular probes

**S protein binding aptamer (SBA) end-labelled molecular probe (10 kbp).** The 10 kbp S protein-targeted molecular probe was prepared by integrating the SBA probe into one of the sticky overhangs of lambda phage DNA (λ-DNA, 48.5 kbp), followed by digesting using a restriction enzyme, Apa I, and separating using agarose gel. In brief, 12.5 μl of commercial λ-DNA (15.8 nM, New England Biolabs) was mixed with 12.5 μl of 1.6 μM of phosphorylated SBA and 25 μl of nuclease-free water (Thermo Scientific) with a λ-DNA-to-SBA ratio of 1:100. A hybridisation procedure was run using a PCR annealing device (TC-3000, TECHNE) with a customised protocol: the mixture was first heated to 75 °C for 5 min, then cooled down gradually to 15 °C at a rate of 1 °C/min, and finally held at 4 °C. SBA sequences SBA can be found in Supplementary Table 1. The resulting product was then ligated by adding 3.6 μl of T4 DNA ligase (400 units/μl, NEB) and 6 μl of 10× T4 ligase reaction buffer (NEB). The ligation was performed at 22 °C for 2 hours, followed by inactivating at 65 °C for 15 min. Finally, 2.5 μl of Apa I (NEB) and 7 μl of 10× rCutSmart buffer (NEB) were added to the resulting mixture and incubated at 25 °C for 30 min, followed by inactivating at 65 °C for 20 min. The resultant product was kept at 4 °C before gel separation.

**N protein binding aptamer (NBA) centre-labelled DNA molecular probe (10 kbp).** The molecular probe targeting N protein was prepared by grafting the NBA probe onto the middle of 10 kbp DNA plasmid (Gene: 318748, ATUM, USA). Briefly, 0.5 μl of 326 nM circular plasmid was first nicked using 2 μl of Nb.BbvCI (10 units/μl, NEB) with the addition of 5 μl of rCutSmart buffer (NEB) and 42.5 μl of nuclease-free water. The mixture was incubated at 37 °C for 60 minutes and inactivated at 80 °C for 20 minutes. This nicking process generated two nicking sites in this plasmid with a distance of 66 bases, which would be displaced by a designed probe comprising the same 66 bases and an extension of NBA sequence (see Supplementary Table 1). The displacement was carried out by performing a hybridisation procedure by mixing the above mixture with 1 μl of 16.3 μM NBA probe to a plasmid-to-NBA ratio of 1:100. The hybridisation procedure was performed according to the protocol described above. The resulting product was then ligated by adding 2 μl of T4 DNA ligase (400 units/μl, NEB) and 5.8 μl of 10× ligation buffer (NEB) using a protocol similar to that described above. Finally, the modified circular plasmid was transferred into a linear state using the restriction enzyme SalI-HF (NEB). To this end, 0.8 μl of SalI-HF (100 units/μl) and 1 μl of additional 10× rCutSmart buffer (NEB) were added into the above mixture and incubated at 37 °C for 60 min, followed by inactivating at 65 °C for 20 min. The obtained NBA-labelled molecular probe was kept at 4 °C prior to gel isolation.

**Preparation of 3-site encoded DNA molecular probe (9.1 kbp) for RNA targeting.** The molecular probe targeting RNA genes was designed and cut from λ-DNA using a customised top-down synthesis approach using a set of commercial nicking and restriction enzymes. DNA probe targeting a specific RNA fragment was encoded at a specific position via displacing a nicking site with designed DNA probes. The 3-site encoded molecular probe (9.1 kbp) was synthesised comprising two nicking steps, a hybridisation for probe displacement, a ligation reaction, and an enzyme digestion step. The preparation procedure is shown in Supplementary Fig. 19.

Briefly, 12.5 μl of λ-DNA was first nicked by mixing with 3.2 μl of Nb.BtsI (10 units/μl, NEB), 5 μl of 10× rCutSmart buffer (NEB), and 29.3 μl of nuclease-free water, followed by incubating at 37 °C for 60 min and inactivating at 80 °C for 20 min. This process created a 45-base breach at the 43.8 kbp position. The λ-DNA was further nicked by adding 3.2 μl of Nt.BsmAI (5 units/μl, NEB) and incubating at 37 °C for 60 min and inactivating at 65 °C for 20 min. This creates another 32-base breach at the 41 kbp position. These two breaches resulting from the nicking reaction together with the sticky overhang, in the end, were then displaced with 3 DNA probes targeting 3 RNA genes areas (i.e., N, S and ORF gene) through a hybridisation reaction according to the protocol mentioned above by adding 1 μl of 100× excess of DNA probes. The resulting product was then ligated by replenishing 3.6 μl of T4 DNA ligase (400 units/μl, NEB) and 6 μl of T4 ligation buffer (NEB)

using the above-mentioned protocol. Finally, the modified λ-DNA was cut using restriction enzyme, PciI (10 units/μl, NEB), at the 39.4 kbp position, generating a 9.1 kbp molecular probe embedded with 3 DNA probes at sites of 1.6 (18%), 4.4 (48%) and 9.1 kbp (100%), Supplementary Fig. 19.

To determine variant mutations, DNA probes that identify the N gene and the mutations in the S gene (D614G, Y144del, G446S, G339D) were also incorporated into a 3-site encoded molecular probe. All the sequences of DNA probes can be found in Supplementary Table 2-8.

**Separation, extraction, and purification of molecular probes.** All above target molecular probes were isolated using an agarose gel electrophoresis procedure. Briefly, the DNA product was firstly mixed with 6× of gel loading dye at a 5:1 (v:v) ratio and loaded onto 0.7% (w%) of agarose gel with up to 0.6 μg of total DNA per well. The gel was then run at a 4 V/cm voltage for 120 min and then transferred into a clean petri dish and stained using 1× SYBR Gold in TBE buffer for around 60 min. By using a gel imaging LED lightbox (Biotium Inc.), the target band was cut down carefully as thin as possible and collected into a small tube. The target molecular probe was then extracted using a commercial gel extraction kit (T1020L, Monarch DNA Gel Extraction Kit, NEB) according to the supplier's protocol. The final concentration of the molecular probe was determined by measuring the UV-Vis absorbance at 260 nm using Nanodrop 2000c (Thermo SCIENTIFIC) (Supplementary Figs. 2, 12, and 19) and stored at −20 °C before use.

**Primer design**
All primers were ordered as DNA from IDT. The primers used for the PCR preamplification of target RNA fragments are designed using NCBI Primer-BLAST[74] with parameters of amplicon size between 70 and 150 nucleotides (nt), primer melting temperatures between 56 and 67 °C, and other parameters at default. Because the double-stranded DNA PCR amplicon would be transcribed to single-stranded RNA before nanopore testing using the HiScribe™ T7 High Yield RNA Synthesis Kit (New England Biolabs), all forward primers were ordered with an upstream T7 promoter sequence (5′-GAAAT TAA TAC GAC TCA CTA TA GGG-3′) in the 5′ end. All the sequences of primers used in this work can be found in Supplementary Table 2-8.

**Preparation of PCR amplicon from full-length synthetic viral RNA**
Full-length synthetic viral RNA of all variants was purchased from Twist Bioscience (California, United States) with an initial concentration of $10^6$ copies per microliter (cp/μl). The RNA samples were aliquoted and stored at -80 °C before use. For the preparation of amplicons used in the nanopore test, the RNA was serially diluted using nuclease-free water (Thermo Scientific) and reverse transcribed into complementary DNA (cDNA) using the iScript cDNA Synthesis Kit (Bio-Rad) by mixing 10 μl RNA, 4 μl 5× reverse-transcription mix, 1 μl reverse transcriptase, and 5 μl of water, followed by 5-min incubation at 25 °C, 20-min incubation at 46 °C, and 1-min inactivation at 95 °C.

The resulting cDNA product was subsequently amplified using a Thermo Scientific™ PCR Master Mix kit. 20 μl of cDNA were mixed with 25 μl of Master Mix buffer (2×), 2.5 μl of forward primers (20 μM each), and 2.5 μl of reverse primers (20 μM each). The mixture was then heated to 95 °C, followed by 35 cycles of 95 °C of denaturation for 30 s, 59 °C of annealing for 30 s, and 72 °C of extension for 30 s. The sample was then kept at 72 °C for 5 min as an extra extension step and finally held at 4 °C.

Before testing under conditions established for the nanopore experiment, the amplicons were transcribed to single-stranded RNA using HiScribe™ T7 High Yield RNA Synthesis Kit (E2040S, New England Biolabs). 5 μl of DNA amplicon products were mixed with 1.5 μl of T7 reaction buffer (10×), 6 μl of NTPs (1.5 μl each), 1.5 μl of T7 RNA polymerase mix, and 1 μl of water to a final volume of 15 μl, followed by

incubating at 37 °C for 2 h. The product was further treated with 1 μl of DNase I (M0303S, New England Biolabs), 2 μl of DNase I reaction buffer, and 2 μl of Nanopure water to remove any remaining DNA by incubating at 37 °C for 10 min and inactivating at 75 °C for 10 min. The PCR products were stored at −80 °C prior to use. The preparation of other gene amplicons with mutations, including D614G, Y144del, G446S, and G339D in S protein, follows the same protocol described above with the replacement of primer pairs specific to those gene areas.

**Lysis of pseudovirus**
The pseudovirus ($10^6$ copies/μl) was generated according to previously published work[75]. Briefly, the S gene, the N gene and ORF1ab were cloned into the CMV/R vector. Using a calcium phosphate transfection kit (Thermo Fisher), psPAX2 vector, pMD2.G vector and CMV/R vector carrying SARS-CoV-2 genes were transfected into $3–8×10^6$ 293 T cells. Pseudoviruses were produced by transfecting cells and subsequently collecting the supernatants after a 48-hour incubation period. The collected supernatants were then filtered using a 0.22 μm syringe filter to ensure purity. Any residual vector DNA present in the samples was subsequently degraded using DNase I. The copy number of pseudoviruses was estimated using RT−qPCR and reference plasmids containing the S gene of SARS-CoV-2. Pseudovirus samples were aliquoted and kept at −80 °C before use. Before measurement, the pseudovirus was inactivated at 65 °C for 30 min and lysed using RIPA lysis buffer (Thermo Fisher) at a volume ratio of 1:1 in the presence of protease inhibitor to release the S protein.

**Clinical sample and ethics statement**
Swabs or saliva were taken from patients or healthy controls in the commercial viral transport medium and inactivated at 65 °C for 30 min in an approved biosafety lab. The viral samples were provided from the Imperial College London testing scheme and were from fully anonymised, redundant samples (i.e., samples left over after testing) and retained for assay development, quality assurance and assay validation. The samples were lysed using commercial lysis buffer and then reverse transcribed into cDNA, as described above. The qPCR was then performed using a SARS-CoV-2 RUO qPCR kit (Cat # 1000637, IDT) according to the supplier's protocol. The sequencing was part of the college's response to ensure that new variants were detected and to detect otherwise unexplained clusters. The consent to providing the sample and to the testing of the sample was provided at test booking through an online process. There was no ethics process for using the viruses obtained from the samples after the testing was complete.

**Preparation of protein from clinical samples**
For protein detection, 20 μl of the patient sample was lysed by incubating with 20 μl of RIPA lysis buffer (Thermo Fisher). The protein released in the lysate was then purified using an Amicon Ultra-0.5 Centrifugal Filter (10 kDa cutoff, Sigma-Aldrich) according to the supplier's protocol for six cycles (15 min per cycle) of ultracentrifuging at 4 °C.

**Preparation of RNA from clinical samples**
For RNA detection, the inactivated nasal/throat swab was first pyrolysed at 95 °C for 5 min to release the viral RNA. The RNA was then reverse transcribed to cDNA using iScript cDNA Synthesis Kit (Bio-Rad). 10 μl of the above lysate was mixed with 4 μl of reverse-transcription mix (5×), 1 μl reverse transcriptase, and 5 μl of water, followed by 5-min incubation at 25 °C, 20-min incubation at 46 °C, and 1-min inactivation at 95 °C.

The resulting cDNA product was subsequently amplified using a Thermo Scientific™ PCR Master Mix kit. 20 μl of cDNA (the product from the above step) was mixed with 25 μl of Master Mix buffer (2×), 2.5 μl of 20 μM forward primers, and 2.5 μl of 20 μM reverse primer. The

primers were designed to cover the regions including mutation of D614G, G339D, and a conservative N gene fragment (see primers in Supplementary Information). The mixture was then heated to 95 °C, followed by 35 cycles of 95 °C of denaturation for 30 s, 59 °C of annealing for 30 s, and 72 °C of extension for 30 s. The sample was then kept at 72 °C for 5 min as an extra extension step and finally held at 4 °C. 3.5 µl of the DNA amplicon products were mixed with 1 µl of T7 reaction buffer (10×), 4 µl of NTPs (1 µl each), and 1.5 µl of T7 RNA polymerase mix (E2040S, New England Biolabs) to a final volume of 10 µl, followed by incubating at 37 °C for 2 h. The resulting product was further treated with 1 µl of DNase I (M0303S, New England Biolabs), 1.5 µl of DNase I reaction buffer, and 2.5 µl of water to remove any remaining DNA by incubating at 37 °C for 10 min and inactivating at 75 °C for 10 min. The PCR products were stored at −80 °C prior to use.

## Nanopore fabrication
All the nanopores were fabricated using a laser-assisted pipette puller (Sutter Instrument, P-2000, USA) by pulling quartz capillaries (GQF100-50-7.5, World Precision Instruments, UK) according to a protocol previously reported by our group[33,34,56] with slightly optimised pulling parameters. Before pulling, the capillaries (inner diameter: 0.5 mm, outer diameter: 1.0 mm, length: 7.5 cm) were treated with oxygen plasma for 30 min using a plasma cleaner (Harrick Plasma). This allows us to remove any organic residues or contaminants on the quartz surface. One capillary was then set up onto the holder of the puller by properly aligning and fixing in the groove. The capillary was then pulled to generate two similar nanopipettes using an optimised two-line pulling parameter: (1) HEAT: 775; FIL: 4; VEL: 30; DEL: 170; PUL: 80, (2) HEAT: 825; FIL: 3; VEL: 20; DEL: 145; PUL: 180. It is worth noting that the pore geometry can be instrument-specific and very sensitive to ambient conditions such as humidity and temperature. In our experimental conditions (30% humidity and 21 °C), the protocol generated nanopores with an average size in the tip of $15 \pm 3$ nm ($n = 5$), based on the SEM estimation and conductance calculation (Supplementary Fig. 3).

## Conductance measurement of nanopipettes
Nanopore conductance measurement was performed before each experiment in 2 M LiCl buffer (with 5 mM MgCl$_2$, 10 mM Tris–HCl, and 1 mM EDTA (pH = 8)) by measuring the current ramping from −400 to +400 mV at a rate of 50 mV. The current–voltage plots were then fitted linearly, and the conductance was estimated from the subtracted slope of the line.

## Translocation experiments
All the translocation experiments through nanopipettes were performed in an electrolyte solution containing 2 M LiCl, 5 mM MgCl$_2$, 10 mM Tris–HCl, and 1 mM EDTA (pH = 8) unless stated otherwise. For the experiments of protein measurements, DNA molecular probes at a final concentration of 200 pM were incubated for approximately 2 h with S or N proteins (Sino Biological) at a specific concentration ratio, as reported in the above-mentioned electrolyte. A nanopipette filled with electrolyte (≈10 µl) was inserted into the above solution (≈200 µl). Two freshly prepared Ag/AgCl electrodes were inserted into the nanopipette and the bath, respectively, acting as working and reference electrodes. A bias, typically 300 mV in this work unless stated otherwise, was applied to drive the translocation of molecules in an outside-to-inside manner. For experiments in human saliva, S and N proteins at the reported concentration were spiked in the saliva (collected from >3 healthy people) and filtered with a 0.22 µm filter to remove any large particles. The saliva was then incubated with 200 pM of aptamer-modified molecular probes at a ratio of 1:20 for 2 h before nanopore measurements.

For protein detection from pseudovirus or patient samples, the lysate or purified proteins were then incubated with SBA and NBA-modified molecular probes (each with 200 pM final concentration) at a volume ratio of 1:50 (for pseudovirus) or 1:10 (for patient samples) for 2 h in 2 M LiCl (5 mM MgCl$_2$, 10 mM Tris–HCl, 1 mM EDTA, pH = 8) at room temperature.

For the detection of RNA fragments, encoded molecular probes at a final concentration of 200 pM were incubated with different concentrations of synthetic RNA (IDT) or RNA amplicons for 2 h. The solution also contained 20 nM of streptavidin and 4 nM of each probe B which is complementary to the second half of the target genomic sequence. To detect RNA genome from clinical samples, 2 µl of RNA product were incubated with 200 µl of 200 pM molecular probes for ≈2 h in 2 M LiCl (5 mM MgCl$_2$, 10 mM Tris–HCl, 1 mM EDTA, pH = 8) at room temperature.

## Data acquisition and analysis
All ionic current recordings were carried out using a high-bandwidth amplifier VC100 (Chimera Instruments). The current data were recorded at a sampling rate of 1 MHz and filtered at 100 kHz. A custom-written application in MATLAB (R2022a), the Nanopore App, was used to analyse the translocation events (see supporting information for further details). Briefly, (1) current-time data was loaded and opened using the Nanopore App. (2) The trace was filtered using a 100 kHz low-pass filter and resampled at 1 MHz. (3) The current baseline was tracked and subtracted. (4) A Poisson distribution was used to determine the open-pore current and thresholds. Typically, a threshold of 7 standard deviations above the mean open-pore current was used to isolate identified events. (5) Events above the threshold were classified as relevant events. (6) Event parameters were saved and exported. (7) After isolating individual events, CUSUM (cumulative sums algorithm)[76,77] was used to fit individual peaks to determine secondary peak amplitude, dwell time, and fractional position. A detailed step-by-step data analysis including justification for fitting parameters and thresholds for each analyte type, can be found in Supplementary Note 1. All reported errors in the manuscript represent one standard deviation.

## Statistics and reproducibility
Number of biological replicates is defined in the legends of the figures. The events statical information, including dwell time, peak current, subpeak normalised peak position, subpeak height, subpeak width, in a single experiment shown as mean ± SD. The statistical data across at least three repeats are shown as mean ± SEM. Statistical analyses were carried out using Matlab (R2022a), Excel (Microsoft Office 365) and OriginLab (2023). The figures were plotted by OriginLab and further imported to Adobe Illustrator (Adobe CC).

## Reporting summary
Further information on research design is available in the Nature Portfolio Reporting Summary linked to this article.

## Data availability
The main data supporting this study's results are available within the paper and its Supplementary Information. Source data are provided with this paper. The example raw traces in this study have been deposited in the Zenodo database at https://zenodo.org/record/8143395. Additional relevant information is available from the corresponding author. Source data are provided with this paper.

## Code availability
The Nanopore App used to analyse the data is available by request to the corresponding authors within a 4-week timeframe.

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

## Acknowledgements

We would like to thank Professor Ruijie Deng for providing the pseudovirus. A.P.I and J.B.E. acknowledge support from BBSRC grant BB/R022429/1, EPSRC grant EP/V049070/1, and Analytical Chemistry Trust Fund grant 600322/05. This project has also received funding from the European Research Council (ERC) under the European Union's Horizon 2020 research and innovation programme (grant agreement Nos. 724300 and 875525). A.E. and P.G. acknowledge support from the Ministry of Education and Science of the Russian Federation, Agreement No. 075-15-2022-264, unique scientific facility reg. No 2512530. R.R. and Y.K. acknowledge support from Japan Society for the Promotion of Science KAKENHI Grants 21H01770, 22K04890, and World Premier International Research Center Initiative (WPI), MEXT, Japan. Research in the P.C. laboratory is supported by the Francis Crick Institute, which receives its core funding from Cancer Research UK (CC2058), the UK Medical Research Council (CC2058), and the Wellcome Trust (CC2058). This research was funded in whole, or in part, by the Wellcome Trust (CC2058). For the purpose of Open Access, the author has applied a CC BY public copyright licence to any Author Accepted Manuscript version arising from this submission. We acknowledge the UK Health Security Agency and St Mary's Hospital for providing clinical samples.

## Author contributions

J.B.E., A.P.I., R.R., S.C., Y.K., and C.E. conceived the idea and designed the experiments. R.R. and S.C. performed the majority of experiments including molecular probe fabrication and nanopore measurement. X.F., Z.Z., Z.L., and W.T. performed the aptamer selection and characterisation. X.W. performed the characterisation of the nanopores. M.D. and C.H. prepared partial molecular probes used in this work. A.R., J.H., N.J.C., and P.C. prepared the viral protein samples for different coronavirus variants. A.B., M.C., P.F., and G.T. extracted and lysised the viruses from patient samples. P.G., A.E., P.N., and A.S. resolved some technical issues encountered during the low-noise single-molecule measurements in this experiment. L.T. and R.R. performed pseudovirus-related experiments. R.R., S.C., J.B.E., and A.P.I. analysed the data. R.R. and S.C. wrote the first draft of the manuscript. All authors contributed to discussions, editing and revision of the manuscript.

## Competing interests

The authors declare no competing interests.
