## [Peer Review File · Nature Communications]

REVIEWER COMMENTS

Reviewer #1 (Remarks to the Author):

The manuscript entitled “Multiplexed detection of viral antigen and RNA using nanopore sensing and encoded molecular probes” by Edel and co-authors presented a single-molecule multiplexed detection of viral antigen and RNA using nanopore sensing and encoded molecular probes. Taking the advantage of nanopore resolution, they can encode the probe in different position on the DNA strand and identify multiple target simultaneously. The sensitivity and selectivity reported is remarkable, in particular, they demonstrated that target RNA fragment with single-base mutation can be distinguished, which allows for the identification of different variants of SARS-CoV-2, including Delta and Omicron. Interestingly, the results from the clinical patient samples show great potential in the practical uses for this method. Overall, this is a well written report with an interesting idea. The supporting information are useful. The cited literature seems appropriate. The quality of figures is good. Therefore, I would like to recommend the publication of this manuscript after a minor revision.

1. This manuscript involved sophisticated data analysis, it would be better for the authors to illustrate in a more detailed way to demonstrate how the data has been extracted and analysed.
2. The author sometimes use ‘wild type’ (Fig. 5), sometimes use ‘wild-type’ (Fig. 7). Please make the whole paper to be consistent.
3. Since the aptamer sequence is new, the authors are recommended to provide the detailed SELEX process in Method section or Supplementary Information.
4. In Supplementary Figure 2c, there’s 8 T linker, however, in the caption, it is 6 T. Please double check and clarify this.
5. It is recommended to give the value of K_d for the SBA in the main text as the NBA does.

Reviewer #2 (Remarks to the Author):

Ren and co-workers report an exciting application of nanopore sensors for amplification-free detection of SARS-CoV-2 proteins (S and N) and transcribed RNA for the S, N, and ORF1b genomic locations. To detect proteins, the authors conjugate long dsDNA to aptamers that specifically bind the protein of interest, and detect it as a secondary peak on the translocation blockage. The peak location within the event reports on the type of construct passing through, so that constructs with aptamers positioned at different locations (center, end, off-center) may be used to multiplex detection. To detect RNA from different gene transcripts, they create secondary peaks using streptavidin bound to a biotinylated oligo

complementary to a specific RNA sequence, while a dangling oligo on the larger DNA construct is complementary to a nearby sequence on that same RNA. Using multiple oligo positions on one construct allows multiplexing. They further show discrimination among different mutant variations using oligos optimized for each task. After checking that their assay detects a pseudovirus as well as synthetic viral RNA, they also test patient samples for wild-type, Delta, and Omicron variants.

On the whole this work is novel, is careful, and is thorough. It is not the first nanopore COVID-detection paper but they do have a great approach that seems robust. I do have some suggestions and questions for the authors, mostly related to choices in data acquisition / analysis / interpretation, but once these are addressed I recommend that this work be published promptly.

Can the authors comment on / explain why their streptavidin tags show such a narrow distribution of fractional peak position? This is surprising given the broad distribution for the N protein and broad distributions observed in other protein-DNA complex nanopore measurements. Is the fractional distribution of the S protein constructs also broad (or has a long tail)? Also, some discussion of how the observed secondary peak amplitudes correspond to the sizes and chemical properties of the various proteins attached to the DNA would be nice. Small point: you may want to mention that it is an S-protein trimer.

The authors cite a fairly large range of nanopore sizes (15nm +/- 3 nm), whose area should vary by more than a factor of 2. How is this range reconciled with the very small range of conductances observed? Were only pores with certain conductances selected for experiments?

It is currently not completely clear how prominent the peak for the N protein is relative to the translocation. From the example events in Figure 3, one is led to believe that these peaks are prominent. However, From S17, it appears that the N peak is below a substantial fraction of the regular (folded?) DNA events. Note that S17 shows the DNA mean at 0.09 nA with a reasonably broad distribution and the N protein peak at 0.1 nA with a very narrow distribution. Similarly, in S30 the N protein clusters well within the regular DNA translocation cluster. This point is related to the questions and suggestions regarding transparency on the peak-finding algorithm mentioned below.

Methods: It is not clear how events are classified as being DNA-only or having a secondary peak. The methods section says only "a multi-step routine was used to identify the secondary peak". The workflow may be the same as for this group's recent JACS paper, but this should be explicitly outlined in this manuscript. I view this as perhaps the only glaring omission in the paper; it would be very useful to understand how events are being classified.

Especially given the bandwidth here, it is likely that many subpeaks are missed. What is the dwell time distribution for the subpeaks (of different varieties)? Their apparent magnitude should also be observed to be reduced at shorter subpeak dwell times; see for example Plesa et al 2013 (<https://pubs.acs.org/doi/full/10.1021/nl3042678>). Are these effects observed?

The authors' statement on data accessibility seems to be that it is not readily accessible; it essentially states that the data shown in the figures is the only data they provide. How big are the raw data files that they could not be hosted on a university server? Could a subset of data, perhaps that used in key figures such as Figs. 2 or 3, be posted? This is not an essential element of the revision, but strikes me as an opportunity for greater transparency.

How reproducible are the results across different nanopores? Is the data in each figure from single pores or aggregated across many? It is not clear whether the blockage levels, for example, are the same in different pores. Figures S5 and S6 have very similar DNA blockages, but S13 shows larger blockages – is that the pore? Are the triplicate-result binding curves all taken in the same pore? If not, are the pores nearly identical in conductance? Throughout the paper, it would be very useful to know whether these experiments are all being performed in just a few, or many, pores, as this is one of the major challenges in the nanopore field.

Do the results from clinical samples (for the RNA, at least) correspond to concentrations that match qRT-PCR results? It may be too late to collect these, but the data is very close to being able to verify the ability to determine viral load. This could be an important and exciting additional result to include.

In a similar vein, relatively little data from the clinical samples is shown in terms of the events and scatterplots characterizing the nanopore data. The pre-clinical samples are described and a bit is shown, but the actual clinical samples only have reported statistics. Showing more of the raw data and events in SI figures would be very informative and interesting to the reader.

The authors are surely aware that many nanopore papers draw questions about the ease of use and requirements for pre-nanopore PCR. I suggest that they carefully re-consider some of the wording in the intro regarding the need for a new diagnostic approach without PCR, fast and on-site, simple, etc, because here they are still using PCR, and it is a highly complex detection scheme.

MINOR COMMENTS and QUESTIONS:

There is substantial prior literature on detection of protein-DNA complexes in nanopores, for example from the groups of Keyser (<https://doi.org/10.1021/ja512521w>), Hall (<https://doi.org/10.1021/nl501340d>), Meller (<https://doi.org/10.1038/srep11643>), Dekker (<https://doi.org/10.1021/acs.nanolett.5b00249>), and Kim (<https://doi.org/10.1021/acs.nano.5b00784>)

with some nice early theoretical work by the group of Aksimentiev (10.1002/elps.201200164). This body of literature should be used to contextualize the approach taken in this work.

What is the source of the pseudovirus? Methods says just “suppliers”.

FIGURES

Figures are generally very nicely laid out and designed. The extensive and thorough SI figures are particularly appreciated, as is the consistent color coding for different assays / samples that is maintained across all figures. A few minor comments and suggestions follow.

Figure 2: what are the protein concentrations for each sample in 2e?

Figure 3 requires scale bars for time and current blockage for panel a. Should show an additional panel similar to 1c. Also, in d, the binning in time is too large and does not show the shape of the distribution.

Fig. 2 and 3: what is the source of the S and N proteins used here?

Are they from the pseudovirus? How are they purified and verified?

Fig 4. A very useful SI figure might be an all-subpeak histogram related to the data split up into the three panels of (iii).

Fig 5: c and d might be better as tables. A and b seem a bit overkill. E is really the most important part of the figure. But, my comment is more about style than substance; OK if authors prefer to leave it.

Across all figures: it is good practice to include the number of events either on the figure or in the caption for histograms and scatter plots. Figures S8 and S9 have this but none of the others seem to.

Reviewer #3 (Remarks to the Author):

This manuscript by Ren et al. reports multiplex detection of SARS-CoV-2 antigens and RNAs using glass pipette-based solid-state nanopores and dsDNA probes. If successfully demonstrated, the potential clinical significance of this technology is clear, and the technology can be translational. However, I am afraid the current manuscript is not suitable for publication due to several major flaws in the experimental design and the data analysis as listed below.

Major Concerns:

In the Introduction section, the authors failed to explain the novelty of this work in comparison to similar previous papers (JACS <https://doi.org/10.1021/jacs.2c13465>; Nature Communications (2021) 12:3515; Nature Nanotechnology (2023) 18:290-298). The first 2 papers are from the same group of authors. It is understandable that the first and the third papers were not cited as they are new, but the second paper was only cited without any detailed explanation. Please clearly explain the novelty of this work.

About DNA folding:

There is partial folding of dsDNA probes, why not optimize the probe sequence or use smaller nanopores to try to avoid folding?

Please double check “However, these secondary peaks, due to carrier folding, could be readily identified by their significantly smaller blockade current of secondary amplitude (90.0 ± 8.6 pA)...” Which structure causes secondary amplitude (90.0 ± 8.6 pA), S protein or folded DNA?

Author claimed to demonstrate “smaller cross-section of folded DNA double-helix compared to the size of the S protein” in Figure S7a, but the figure only shows 3D structure of S protein without dimension. It is the authors responsibility to demonstrate a clear comparison of dimensions of folded DNA vs. S protein.

Related to the previous comments, even with a comparison of dimensions, it is still hardly a direct evidence that the partially elevated peak is due to partially folded DNA probe. An experimental comparison between signals from a known folded dsDNA and a known unfolded dsDNA may help.

About multiplex gene detection:

Figure 4e shows only one calibration curve. Is this curve used for all 3 genes? It seems, in Figure 4d, that responses to different genes at the same concentration are different. In addition, how are the SD (error bars) calculated if 3 genes are sharing the same curve?

Author mentioned that “For all three genes, 35-cycle PCR amplification was performed...” After reviewing the Method section, it seems RT-PCR was done to amplify the target genes in clinical samples before nanopore measurement. In this case, the nanopore experiments seems redundant, as these target genes could be directly quantified by RT-qPCR, which combines amplification and quantitative

detection into one step. Even though the author argues that their method has “approximately two orders of magnitude lower LOD than gold standard RT-qPCR”, clinical samples tested were all confirmed RT-qPCR positives. Without testing patients with false negative RT-qPCR, the benefit of adding many complex steps for nanopore measurement is not clear.

About preclinical test in saliva:

Author stated that “S and N protein was spiked in the saliva (with a final concentration of 20 nM)”. How is the concentration calculated? in what solution/matrix?

Author stated that “According to the calibration curve in Fig. 2d, we estimated the concentration of S protein to be approximately 0.5 pM, Supplementary Fig. S31.” From the text related to Figure 2d, the calibration curve was not established using the same solution used in the Pseudovirus experiment, which includes “lysis and extraction buffer” etc. The calibration curve in Figure 2d cannot be used here. In addition, author only calculated LOD as 3σ above the background but did not calculate LOQ (limit of quantification). If LOQ is greater than 0.5pM, then the assay cannot quantify at 0.5pM.

Again, Figure 6b results are based on a calibration curve established in a less complex solution (Figure 4e). It appears that Figure 6av is the same as Figure 2d and 3b; Figure 6bv is the same as Figure 4e. The same data should not be presented twice with different figure number references. This is not acceptable.

About clinical test:

The ethics statement is inadequate.

Details about how each patient sample was PCR confirmed are needed.

Again, clinical samples are tested in a different buffer/matrix environment. Fortunately, there is no concentration calculation in this section, only binding ratio was presented.

It is premature to conclude that the test can detect mutations reliably with only 5 clinical samples per mutation. The authors should make this clear.

In the Conclusion section, although the multiplex detection capacity is plausible, current results do not support the claim of “improved accuracy and ultra-high sensitivity”. There is no comparison between this assay and benchmark methods, so improved accuracy cannot be claimed. Neither analytical sensitivity nor clinical sensitivity can be rigorously evaluated in this study. Analytical sensitivity should be assessed by calibration curves established using the same buffer as in real-world applications. Clinical sensitivity needs to be assessed using more samples and, more importantly, in parallel with clinical test results and patient information that can help reliably analyze the results (e.g. sample collection time, treatment time, symptoms, etc.)

Minor Comments:

The author claimed that “The binding ratio for S protein increased initially and plateaued after 20 nM.” However, in Figure 2d, the binding ratio of 20 nM and 200 nM showed increase.

Figure S12b should be N protein?

According to Figure 4b(i), in Figure 4c(i) the N site is positive, but the legend states negative. Similar inconsistency is also seen in Figure 4c(ii).

What is the Z axis title and unit in Figure 4d?

Figure 5e is confusing. I suggest matching the target of each site (colored blocks on the top) to its fractional position. For example, N at last etc.

Figure S30 indicates ~9% events are short dwell time events suspected to be induced by background molecules. However, in the first 200 events acquired from saliva indicated in Figure S29, none of them were from background molecules. This is highly unlikely to happen statistically.

“We also demonstrated sensitivity down to 1 copy of RNA per 100 μ l via a preamplified step in nasal/throat swab patient samples.” sounds confusing. It suggests 1 copy/100 μ l sensitivity without the PCR amplification which is not the case.

Please double check all references. Reference numbers are not appearing in the correct order in the text.

REVIEWER Response

Reviewer #1 (Remarks to the Author):

The manuscript entitled “Multiplexed detection of viral antigen and RNA using nanopore sensing and encoded molecular probes” by Edel and co-authors presented a single-molecule multiplexed detection of viral antigen and RNA using nanopore sensing and encoded molecular probes. Taking the advantage of nanopore resolution, they can encode the probe in different position on the DNA strand and identify multiple target simultaneously. The sensitivity and selectivity reported is remarkable, in particular, they demonstrated that target RNA fragment with single-base mutation can be distinguished, which allows for the identification of different variants of SARS-CoV-2, including Delta and Omicron. Interestingly, the results from the clinical patient samples show great potential in the practical uses for this method. Overall, this is a well written report with an interesting idea. The supporting information are useful. The cited literature seems appropriate. The quality of figures is good. Therefore, I would like to recommend the publication of this manuscript after a minor revision.

1. This manuscript involved sophisticated data analysis, it would be better for the authors to illustrate in a more detailed way to demonstrate how the data has been extracted and analysed.

We have added a section in supporting information about data analysis explaining the analysis procedure.

2. The author sometimes use ‘wild type’ (Fig. 5), sometimes use ‘wild-type’ (Fig. 7). Please make the whole paper to be consistent.

We have revised the manuscript. All ‘wild type’ has been changed to ‘wild-type’

3. Since the aptamer sequence is new, the authors are recommended to provide the detailed SELEX process in Method section or Supplementary Information.

Thanks for the good suggestion. We have included a detailed description of the SELEX process in the Supplementary Information.

4. In Supplementary Figure 2c, there’s 8 T linker, however, in the caption, it is 6 T. Please double check and clarify this.

The supplementary Figure 2 caption has been revised.

5. It is recommended to give the value of K_d for the SBA in the main text as the NBA does.

We calculated the values as K_d for NBA = 0.73 nM K_d for SBA = 9.86 nM

We have updated Fig.S1 and Fig. S10 captions with the K_d value.

Reviewer #2 (Remarks to the Author):

Ren and co-workers report an exciting application of nanopore sensors for amplification-free detection of SARS-CoV-2 proteins (S and N) and transcribed RNA for the S, N, and ORF1b genomic locations. To detect proteins, the authors conjugate long dsDNA to aptamers that specifically bind the protein of interest, and detect it as a secondary peak on the translocation blockage. The peak location within the event reports on the type of construct passing through, so that constructs with aptamers positioned at different locations (center, end, off-center) may be used to multiplex detection. To detect RNA from different gene transcripts, they create secondary peaks using streptavidin bound to a biotinylated oligo complementary to a specific RNA sequence, while a dangling oligo on the larger DNA construct is complementary to a nearby sequence on that same RNA. Using multiple oligo positions on one construct allows multiplexing. They further show discrimination among different mutant variations using oligos optimized for each task. After checking that their assay detects a pseudovirus as well as synthetic viral RNA, they also test patient samples for wild-type, Delta, and Omicron variants.

On the whole this work is novel, is careful, and is thorough. It is not the first nanopore COVID-detection paper but they do have a great approach that seems robust. I do have some suggestions and questions for the authors, mostly related to choices in data acquisition / analysis / interpretation, but once these are addressed I recommend that this work be published promptly.

Can the authors comment on / explain why their streptavidin tags show such a narrow distribution of fractional peak position? This is surprising given the broad distribution for the N protein and broad distributions observed in other protein-DNA complex nanopore measurements. Is the fractional distribution of the S protein constructs also broad (or has a long tail)? Also, some discussion of how the observed secondary peak amplitudes correspond to the sizes and chemical properties of the various proteins attached to the DNA would be nice. Small point: you may want to mention that it is an S-protein trimer.

For N protein and S protein detection, as it is straightforward to differentiate the sub-peak in the middle or in the end, we took all the translocation events (including folding and unfolding) into account to ensure the accuracy for binding ratio. As the folding of DNA carrier would shift the fractional position within the whole event (as shown in the example below); therefore, the distribution for N protein fractional position is broader. However, for the streptavidin binding experiment, we need to differentiate the 0.2 and 0.5 point, which is used to measure the mutants. We discarded the folding events for the statistics as it would affect the accuracy of determining the fractional positions of a given target. Therefore, the fractional peak position shows narrower as it should be.

We have addressed the comparison of secondary peak amplitudes for the S and N proteins in the main text on Page 7: "These secondary peaks exhibit an average amplitude of 85.2 ± 23.8 pA and a dwell time of 25 ± 18 μ s. It is worth mentioning that a fraction of the observed signal events corresponds to partial folding, both in the absence and presence of S protein (Supplementary Fig. S4). However, these secondary peaks resulting from probe folding can be easily distinguished by their significantly smaller blockade current of 32.7 ± 9.9 pA, as shown in Fig. 2c and Supplementary Fig. S5-6. This smaller current can be attributed to the smaller width of the folded DNA double-helix (compared to the size of the S protein, (see Supplementary Fig. S7a for the dimensions of S protein)).".

We are using the S1 domain of S protein, not S-protein trimer, we have now clarified this in the main text.

The authors cite a fairly large range of nanopore sizes (15nm +/- 3 nm), whose area should vary by more than a factor of 2. How is this range reconciled with the very small range of conductances observed? Were only pores with certain conductances selected for experiments?

The nanopore size range is calculated from SEM images however we use nanopore conductance measurements and peak current SNR for the carrier molecule as indicator of nanopore device-to-device variation. As such we try to use nanopores with minimal conductance deviation (less than 4.6%) in the same set of experiments (for example for S protein detection) to ensure the signals are comparable, see Supplementary Fig. 3; but it should be noted that the pore size could also be different. We have added one section in supporting information about using IV to calculate the pore size.

It is currently not completely clear how prominent the peak for the N protein is relative to the translocation. From the example events in Figure 3, one is led to believe that these peaks are prominent. However, From S17, it appears that the N peak is below a substantial fraction of the regular (folded?) DNA events. Note that S17 shows the DNA mean at 0.09 nA with a reasonably broad distribution and the N protein peak at 0.1 nA with a very narrow distribution. Similarly, in S30 the N protein clusters well within the regular DNA translocation cluster. This point is related to the questions and suggestions regarding transparency on the peak-finding algorithm mentioned below.

The width or height of the subpeaks does not impact the conclusions drawn from the data because we rely on the ratio of bound to unbound events to distinguish positive and negative events. For the height of N protein signal, it is within the DNA translocation cluster because the size of N protein is small. In Figure S17, since the number of detected N protein events ($n = 60$) is significantly lower than that of the unbound DNA carrier ($n = 1877$), the distribution of N protein numbers appears narrower compared to that of the DNA carrier (folded and unfolded). This pattern is also observed in Figure S30. This circumstance actually motivated us to design the N protein DNA carrier with the binding site in the middle instead of at the end, as it would have been challenging to differentiate the protein binding signal from the DNA folding signal. We acknowledge the need to provide more details on the peak selection in our data analysis. To address this, we have included a data analysis section in the Supplementary Information and have made some raw data available for reference.

Methods: It is not clear how events are classified as being DNA-only or having a secondary peak. The methods section says only "a multi-step routine was used to identify the secondary peak". The workflow may be the same as for this group's recent JACS paper, but this should be explicitly outlined in this manuscript. I view this as perhaps the only glaring omission in the paper; it would be very useful to understand how events are being classified.

Please refer to Reviewer 1, Comment 1. We have added one section for data analysis in Supplementary Information.

Especially given the bandwidth here, it is likely that many subpeaks are missed. What is the dwell time distribution for the subpeaks (of different varieties)? Their apparent magnitude should also be observed to be reduced at shorter subpeak dwell times; see for example Plesa et al 2013 (<https://pubs.acs.org/doi/full/10.1021/nl3042678>). Are these effects observed?

For the recording we used Chimera VC100 amplifier and all data was recorded at 4Ms per second and subsequently filtered using a low-pass digital filter. Through careful evaluation of events under different filters, we ultimately chose a 1 microsecond sampling rate and a 30 kHz low-pass filter to resample and refilter the data. This selection was made to ensure a high signal-to-noise ratio while minimizing the loss of subpeaks.

In our manuscript, we did not extensively discuss the subpeak dwell time because it falls outside the focus of this particular study and introduces additional complexity. However, we sincerely appreciate the reviewer for bringing it up, as it provides an opportunity for discussion. Firstly, we would like to highlight that the phenomenon observed by Plesa et al. pertains to unbound proteins, which differs from our specific case. In our work, the translocation speed of bound proteins is primarily influenced by the DNA carrier rather than the proteins themselves. Interestingly, we do observe some phenomena that align with the reviewer's comments.

SubPeakHeight = 0.079096 nA
SubPeakWidth = 0.06624 ms
FractionalPos = 0.18555
FractionalWidth = 0.061717
RelSubpeak = 2.4146

SubPeakHeight = 0.060444 nA
SubPeakWidth = 0.02784 ms
FractionalPos = 0.8117
FractionalWidth = 0.026581
RelSubpeak = 2.1379

For instance, when considering the same binding position (0.2), we notice differences in subpeak dwell time and peak current depending on the entry direction. The first entry direction exhibits a longer subpeak dwell time and a higher peak current, while the second entry direction shows a shorter and smaller subpeak. This discrepancy likely arises due to the non-uniform motion of DNA molecules. Similar phenomena have been reported in Bell et al. *Nature Communications volume 8, Article number: 380 (2017)*. However, due to differences in the experimental environment, we cannot directly apply their model to explain our results. Nevertheless, we believe this aspect would be intriguing to explore further in future studies.

The authors' statement on data accessibility seems to be that it is not readily accessible; it essentially

states that the data shown in the figures is the only data they provide. How big are the raw data files that they could not be hosted on a university server? Could a subset of data, perhaps that used in key figures such as Figs. 2 or 3, be posted? This is not an essential element of the revision, but strikes me as an opportunity for greater transparency.

We agree with the reviewer and source data will uploaded as part of the resubmission.

How reproducible are the results across different nanopores? Is the data in each figure from single pores or aggregated across many? It is not clear whether the blockage levels, for example, are the same in different pores. Figures S5 and S6 have very similar DNA blockages, but S13 shows larger blockages – is that the pore? Are the triplicate-result binding curves all taken in the same pore? If not, are the pores nearly identical in conductance? Throughout the paper, it would be very useful to know whether these experiments are all being performed in just a few, or many, pores, as this is one of the major challenges in the nanopore field.

It should be noted that the data presented in typical traces, bar charts, or scatter plots are collected from the same nanopore. When error bars are shown, they are typically obtained from at least three different nanopores. We do not combine data across multiple pores.

The dimensions of the nanopore can vary due to pulling ambient conditions such as temperature and humidity. However, within the same set of experiments, we make an effort to use nanopores with conductance values as similar as possible to minimize variation. The experiments depicted in Figure S5/S6, which pertain to S protein detection, were conducted using nanopores pulled during the same period. On the other hand, the nanopores employed in the experiments for Figure S13 might differ since that batch was pulled at a different time. Nevertheless, our approach is specifically designed to detect the binding of target proteins by analysing sub-peaks at different fractional positions, rather than relying on peak height. This strategy effectively addresses the issue of nanopore variations.

Do the results from clinical samples (for the RNA, at least) correspond to concentrations that match qRT-PCR results? It may be too late to collect these, but the data is very close to being able to verify the ability to determine viral load. This could be an important and exciting additional result to include. In a similar vein, relatively little data from the clinical samples is shown in terms of the events and scatterplots characterizing the nanopore data. The pre-clinical samples are described and a bit is shown, but the actual clinical samples only have reported statistics. Showing more of the raw data and events in SI figures would be very informative and interesting to the reader.

We compared our nanopore data with qRT-PCR results for the Delta and Omicron patient samples see (Supplementary Table 9). Unfortunately the wildtype data RT-qPCR data was not available at the time of collection. We have added some example traces and events for the clinical data, in Supporting information (Supplementary Fig.34).

The authors are surely aware that many nanopore papers draw questions about the ease of use and requirements for pre-nanopore PCR. I suggest that they carefully re-consider some of the wording in

the intro regarding the need for a new diagnostic approach without PCR, fast and on-site, simple, etc, because here they are still using PCR, and it is a highly complex detection scheme.

We agree with the reviewer's suggestion. We have made appropriate changes in the Introduction as well as the Conclusions sections.

MINOR COMMENTS and QUESTIONS:

There is substantial prior literature on detection of protein-DNA complexes in nanopores, for example from the groups of Keyser (<https://doi.org/10.1021/ja512521w>), Hall (<https://doi.org/10.1021/nl501340d>), Meller (<https://doi.org/10.1038/srep11643>), Dekker (<https://doi.org/10.1021/acs.nanolett.5b00249>), and Kim (<https://doi.org/10.1021/acsnano.5b00784>) with some nice early theoretical work by the group of Aksimentiev (10.1002/elps.201200164). This body of literature should be used to contextualize the approach taken in this work.

We added these references and discussion in the Introduction (paragraph 5: line 5-7).

What is the source of the pseudovirus? Methods says just "suppliers".

pseudovirus of SARS-CoV-2 was provided by Professor Ruijie Deng's lab generated using the lentiviral vector system according to their recent published work (*Nat. Biomed. Eng* 6, 957–967 (2022)). We have referenced this in the text.

FIGURES

Figures are generally very nicely laid out and designed. The extensive and thorough SI figures are particularly appreciated, as is the consistent color coding for different assays / samples that is maintained across all figures. A few minor comments and suggestions follow.

Figure 2: what are the protein concentrations for each sample in 2e?

The S protein concentration is 200 nM for each virus type. We added this in the legend. We also added the N protein concentration (20 nM) in the Fig. 3.

Figure 3 requires scale bars for time and current blockage for panel a. Should show an additional panel similar to 1c. Also, in d, the binning in time is too large and does not show the shape of the distribution.

The scale bars for time and current blockage have been added. We also re-binned the time data in panel d to show the shape of distribution.

Fig. 2 and 3: what is the source of the S and N proteins used here?
Are they from the pseudovirus? How are they purified and verified?

The S and N proteins used in Fig. 2 and 3 are not from pseudovirus but from the commercial supplier (Sino Biological). This was mentioned in the Methods section - 'Translocation experiments'.

Fig 4. A very useful SI figure might be an all-subpeak histogram related to the data split up into the three panels of (iii).

We have added a new SI figure to support this. (Supplementary Fig. 35?)

Fig 5: c and d might be better as tables. A and b seem a bit overkill. E is really the most important part of the figure. But, my comment is more about style than substance; OK if authors prefer to leave it.

We made several iterations of this figure and concluded that including tables was necessary. Without tables, the panels in (e) would be challenging to comprehend. Additionally, we made some adjustments to panel (e) to ensure that the mutant point aligns with the statistics presented in the bar chart, thereby enhancing the clarity of this figure.

Across all figures: it is good practice to include the number of events either on the figure or in the caption for histograms and scatter plots. Figures S8 and S9 have this but none of the others seem to.

The manuscript has been updated accordingly.

Reviewer #3 (Remarks to the Author):

This manuscript by Ren et al. reports multiplex detection of SARS-CoV-2 antigens and RNAs using glass pipette-based solid-state nanopores and dsDNA probes. If successfully demonstrated, the potential clinical significance of this technology is clear, and the technology can be translational. However, I am afraid the current manuscript is not suitable for publication due to several major flaws in the experimental design and the data analysis as listed below.

Major Concerns:

In the Introduction section, the authors failed to explain the novelty of this work in comparison to similar previous papers (JACS <https://doi.org/10.1021/jacs.2c13465>; Nature Communications (2021) 12:3515; Nature Nanotechnology (2023) 18:290-298). The first 2 papers are from the same group of authors. It is understandable that the first and the third papers were not cited as they are new, but the second paper was only cited without any detailed explanation. Please clearly explain the novelty of this work.

The JACS paper we published is not entirely relevant to this discussion as it explores a somewhat different topic. It focuses on the use of supercharged polypeptides as molecular carriers for nanopore detection.

In Nature Communications (2021) 12:3515, we combined single-molecule fluorescence with nanopore sensing to enable multiplexed detection. However, this approach is substantially different as it requires fluorescence labelling and an optical setup. We have also addressed this discussion in the Introduction section, specifically in Paragraph 5, lines 8-10.

Nature Nanotechnology (2023) 18:290-298, was not yet published at the time of our manuscript submission. However, their work primarily focuses on sensing viral RNA, whereas our study demonstrates the simultaneous detection of viral antigens and RNA sequences. We have included this reference in the related discussion in the Introduction section, specifically in Paragraph 5, lines 11-13.

About DNA folding:

There is partial folding of dsDNA probes, why not optimize the probe sequence or use smaller nanopores to try to avoid folding?

We acknowledge that optimizing the probe sequences or using smaller nanopores could potentially decrease the occurrence of DNA translocation folding events. However, we believe it would be challenging to completely eliminate all folding events by optimizing the DNA probes alone. Additionally, although employing smaller pores can reduce folding events, it would also hinder the translocation of protein-bound DNA carriers since proteins are typically larger than folded DNA.

Nonetheless, these folding events do not impact the results or their interpretability, as we have implemented a customized code to select the secondary peaks corresponding to the binding of the target, distinguishing them from DNA folding events. A newly-added section in the Supplementary Information outlines how we analyse the data and select these bound events. We believe this approach provides a more convenient and straightforward solution to address this issue.

Please double check “However, these secondary peaks, due to carrier folding, could be readily identified by their significantly smaller blockade current of secondary amplitude (90.0 ± 8.6 pA)...”

Which structure causes secondary amplitude (90.0 ± 8.6 pA), S protein or folded DNA?

Author claimed to demonstrate “smaller cross-section of folded DNA double-helix compared to the size of the S protein” in Figure S7a, but the figure only shows 3D structure of S protein without dimension. It is the authors responsibility to demonstrate a clear comparison of dimensions of folded DNA vs. S protein.

Related to the previous comments, even with a comparison of dimensions, it is still hardly a direct evidence that the partially elevated peak is due to partially folded DNA probe. An experimental comparison between signals from a known folded dsDNA and a known unfolded dsDNA may help.

The translocation of folded DNA leads to the appearance of a secondary peak at either the beginning or the end, as demonstrated by the translocation of DNA carriers shown in Supplementary Fig. S5, S13 and S20. This phenomenon has also been documented in previous studies conducted by other research groups (*Nano Lett.* 10, 2493–2497 (2010); *Phys. Rev. E*, 2005, 71, 051903; *Nano Lett.* 2010, 10, 8, 3163–3167; *Nat Commun* 10, 4473 (2019)).

The S protein's size, as determined by CryoEM in prior research (*Cell*, 2020, 180,281–292; *Science*, 367, 1260–1263 (2020)), is approximately 16 nm, which is significantly larger than the cross-section of the folded DNA strand. The binding of the protein to the DNA and their subsequent translocation result in a larger secondary peak compared to those induced by DNA folding. To distinguish between these signals, we developed a customized code that identifies the target signals. This code is detailed in the Supplementary Information.

To make this comparison clear, we have compared the S protein subpeak and DNA folding level peak rather than compare the whole peak amplitude: “These secondary peaks exhibit an average amplitude of 85.2 ± 23.8 pA and a dwell time of 25 ± 18 μ s. It is worth mentioning that a fraction of the observed signal events corresponds to partial folding, both in the absence and presence of S protein (Supplementary Fig. S4). However, these secondary peaks resulting from probe folding can be easily distinguished by their significantly smaller blockade current of 32.7 ± 9.9 pA, as shown in Figure 2c and Supplementary Fig. S5-6. This smaller current can be attributed to the smaller width of the folded DNA double-helix (compared to the size of the S protein, (see Supplementary Fig. S7a for the dimensions of S protein). Additionally, an experimental control was conducted using S protein alone,

without the presence of molecular probes, resulting in minimal translocation events (Supplementary Fig. S7b).”

About multiplex gene detection:

Figure 4e shows only one calibration curve. Is this curve used for all 3 genes? It seems, in Figure 4d, that responses to different genes at the same concentration are different. In addition, how are the SD (error bars) calculated if 3 genes are sharing the same curve?

In this case, the calibration curve was constructed solely based on the response of the N gene binding ratio at a fractional position of 1.0, rather than considering all three genes. The other two binding sites (0.2 and 0.5) were designed primarily to verify the presence or absence of specific genes or mutants in the subsequent study (as depicted in Fig. 5), rather than for quantitative analysis in this particular context. We have explicitly clarified this information in the caption of Figure 4 to prevent any confusion.

Author mentioned that “For all three genes, 35-cycle PCR amplification was performed...” After reviewing the Method section, it seems RT-PCR was done to amplify the target genes in clinical samples before nanopore measurement. In this case, the nanopore experiments seems redundant, as these target genes could be directly quantified by RT-qPCR, which combines amplification and quantitative detection into one step. Even though the author argues that their method has “approximately two orders of magnitude lower LOD than gold standard RT-qPCR”, clinical samples tested were all confirmed RT-qPCR positives. Without testing patients with false negative RT-qPCR, the benefit of adding many complex steps for nanopore measurement is not clear.

The integration of PCR with nanopore detection was primarily motivated by two key considerations. Firstly, the intricate secondary structure of long RNA molecules (approximately 30 kb) poses challenges for probes to access and bind to specific regions of interest. Secondly, in many real-world scenarios, unprocessed RNA concentrations can be relatively low, typically in the range of 10^{-18} M (*Nat. Biomed. Eng* 6, 968–978 (2022)). Given that PCR is a well-established technique, we opted to combine it with nanopore detection in this study to demonstrate the proof-of-concept. However, simpler approaches like LAMP and RCA could also be employed as alternatives.

Nonetheless, the primary advantage of combining these techniques with nanopore-based detection lies in the enhanced multiplexing capabilities that allows the combined detection of multiple proteins and nucleic acids from the sample. This enables the simultaneous detection of multiple targets and the differentiation of key mutations within a single test, eliminating the need for DNA sequencing.

About preclinical test in saliva:

Author stated that “S and N protein was spiked in the saliva (with a final concentration of 20 nM)”. How is the concentration calculated? in what solution/matrix?

Author stated that “According to the calibration curve in Fig. 2d, we estimated the concentration of S protein to be approximately 0.5 pM, Supplementary Fig. S31.” From the text related to Figure 2d, the calibration curve was not established using the same solution used in the Pseudovirus experiment, which includes “lysis and extraction buffer” etc. The calibration curve in Figure 2d cannot be used here. In addition, author only calculated LOD as 3σ above the background but did not calculate LOQ

(limit of quantification). If LOQ is greater than 0.5pM, then the assay cannot quantify at 0.5pM. Again, Figure 6b results are based on a calibration curve established in a less complex solution (Figure 4e). It appears that Figure 6av is the same as Figure 2d and 3b; Figure 6bv is the same as Figure 4e. The same data should not be presented twice with different figure number references. This is not acceptable.

As described in the "Translocation experiments" section of the Methods, the pseudovirus lysate was incubated with SBA and NBA-DNA molecular probes in a buffer at a volume ratio of 1:50, equivalent to a 2% content in the buffer. For the detection of RNA in synthetic samples, only 2 μ L of the PCR products were added to 200 μ L of detection buffer during the validation of this method. While not identical, we believe that the 1% or 2% matrix content in the buffer does not significantly impact the detection environment. Therefore, we utilized calibration curves obtained from buffer-only experiments to estimate the abundance of target proteins or RNA in the pseudovirus or synthetic samples.

We appreciate the reviewer's concern regarding the limit of quantification (LOQ), and we have addressed this by conducting LOQ analysis for all calibration curves (Fig. 2d, 3b, and 4e). The LOQ values were calculated using a threshold of 10σ above the background signal. We have also added this information in the corresponding text to provide additional clarity.

About clinical test:

The ethics statement is inadequate.

Details about how each patient sample was PCR confirmed are needed.

Again, clinical samples are tested in a different buffer/matrix environment. Fortunately, there is no concentration calculation in this section, only binding ratio was presented.

It is premature to conclude that the test can detect mutations reliably with only 5 clinical samples per mutation. The authors should make this clear.

We have revised the ethics statement for the use of clinical samples, providing specific details on how patient samples were confirmed. Please refer to the Methods section, specifically the Clinical Sample and Ethics Statement subsection, for the updated information.

Regarding the detection of mutations, we acknowledge that drawing reliable conclusions based on only five clinical samples would be premature. As a result, we have modified the wording of the clinical results to reflect this limitation and emphasized the need for additional clinical trials to further investigate this aspect.

In the Conclusion section, although the multiplex detection capacity is plausible, current results do not support the claim of "improved accuracy and ultra-high sensitivity". There is no comparison between this assay and benchmark methods, so improved accuracy cannot be claimed. Neither analytical sensitivity nor clinical sensitivity can be rigorously evaluated in this study. Analytical sensitivity should be assessed by calibration curves established using the same buffer as in real-world applications. Clinical sensitivity needs to be assessed using more samples and, more importantly, in parallel with clinical test results and patient information that can help reliably analyze the results (e.g. sample collection time, treatment time, symptoms, etc.)

This manuscript presents a proof-of-concept demonstration of the use of nanopore technology combined with designed DNA molecular probes for multiplexed detection of viral antigens and RNA.

While our findings are promising, we recognize that there is still a considerable distance to cover before this method can be implemented in practical clinical settings. It requires extensive efforts to validate its performance in real-world scenarios. Consequently, we have revised the conclusion to reflect this perspective. The clinical claim has been downplayed, and we have explicitly stated that further clinical trials and validation are necessary before practical applications can be realised.

Minor Comments:

The author claimed that “The binding ratio for S protein increased initially and plateaued after 20 nM.” However, in Figure 2d, the binding ratio of 20 nM and 200 nM showed increase. Figure S12b should be N protein?

It's N protein, Figure S12b has been revised.

According to Figure 4b(i), in Figure 4c(i) the N site is positive, but the legend states negative. Similar inconsistency is also seen in Figure 4c(ii).

Figure. 4 has been revised.

What is the Z axis title and unit in Figure 4d?

It's count, which has been added into the figure.

Figure 5e is confusing. I suggest matching the target of each site (colored blocks on the top) to its fractional position. For example, N at last etc.

We have revised the figure to make it easier to understand.

Figure S30 indicates ~9% events are short dwell time events suspected to be induced by background molecules. However, in the first 200 events acquired from saliva indicated in Figure S29, none of them were from background molecules. This is highly unlikely to happen statistically.

In the first 200 events, we do have several short events, however they cannot be used to calculate the binding ratio so we removed them.

“We also demonstrated sensitivity down to 1 copy of RNA per 100 μ l via a preamplified step in nasal/throat swab patient samples.” sounds confusing. It suggests 1 copy/100 μ l sensitivity without the PCR amplification which is not the case.

Apologies for the confusion. We have reworded as “We also demonstrated sensitivity down to 1 copy of RNA per 100 μ l with a 35-cycle PCR amplified step in nasal/throat swab patient samples.”

Please double check all references. Reference numbers are not appearing in the correct order in the text.

Thanks. References and its numbers have been checked.

REVIEWER COMMENTS

Reviewer #1 (Remarks to the Author):

The authors have addressed my comments very well, so I'd like to recommend the publication in Nature Communications.

Reviewer #2 (Remarks to the Author):

The authors are to be commended on many thorough and careful additions and revisions to this manuscript in response to reviewer feedback. However, one major issue that requires further clarification is the data analysis:

While it's now clear that the authors used their Nanopore App for analysis, the procedure for identifying secondary peaks is still completely elided: "a multi-step routine was used to identify the secondary peak". What are the multiple steps? Are they different for different types of secondary peaks? Please include a description of secondary peak identification analysis in the Methods section; this entire paper is about the subpeaks, so the method for statistically identifying and characterizing them should be explained thoroughly and clearly.

Note that in the SI Note "Data Analysis", the analysis step #s do not line up with the #s in the main text (example: main text analysis step 2 is "Track and subtract the baseline" while SI Note step 2 is "Resample and Filter Trace").

Moreover, in the SI Note "Data Analysis" the only mention that is made of sub-peak finding is currently in SI step #6 (the SECOND SI step #6, "Subpeak Analysis") and says just that "By clicking the 'Auto Subpeaks' analysis button, the subpeak details for all events containing subpeaks will be compiled and listed. Thresholds are employed to distinguish genuine positive events from potential false positives caused by folding signals." This does not explain how the analysis is done - what thresholds and statistical tests are used to identify and screen potential subpeaks.

Is the figure on p. 46 of the SI (no caption) intended to be an additional SI figure related to the subpeak finding? If so, it needs a caption and explanation, and a clear reference within the SI note. It appears to show examples of subpeaks and folds rather than explaining their statistical determination. And, given

that it is showing a data set in which only 7 events have subpeaks, what is the reader meant to learn from the histograms of those seven events into five bins ("subpeak current")?

On a related note, the authors' assertion in the response to reviewers that "The width or height of the subpeaks does not impact the conclusions drawn from the data because we rely on the ratio of bound to unbound events to distinguish positive and negative events" does not make sense since they (presumably; methods unclear) rely on width and height of subpeaks to differentiate folded states from bound protein subpeaks.

The authors also state that they "have made some raw data available for reference" but I do not see this in the review materials - only figure data files of processed data.

Reviewer #3 (Remarks to the Author):

Regrettably, some of the concerns are not successfully addressed. In general, the authors answered most questions only in the response letter, but failed to incorporate the explanations into the paper to improve scientific rigor.

The authors explained the issues of partial folding of DNA probe sufficiently. Please implement this explanation in the manuscript so the readers may understand why additional optimization to eliminate folding is not necessary as it does not impact result in interpretability.

In the multiplex gene detection section, the current text implies simultaneous quantification which is not achieved. Please clarified, in the main text, that the calibration curve was constructed for N gene, and that the other two binding sites were used to verify the presence of the other 2 genes. This is only vaguely mentioned in the figure caption in the current manuscript.

Please add the rationale of the need of PCR to the manuscript. This is only explained in the response letter but not incorporated into the paper.

Please acknowledge the difference between the buffers/matrices used for calibration curve and the other experiments in the manuscript. The lack of this information will cause confusion and reproducibility for potential readers.

There is no response to this concern: "It appears that Figure 6av is the same as Figure 2d and 3b; Figure 6bv is the same as Figure 4e. The same data should not be presented twice with different figure number references." The repetitive figures need to be removed.

The ethics statement is still inadequate. The most important information is the IRB approval. Whether this is a human study or not (only using redundant samples as materials without identifying information) should be determined by an Institutional Review Board or an equivalent committee, and the approved project number should be included in the ethic statement.

REVIEWER Response

Reviewer #2 (Remarks to the Author):

The authors are to be commended on many thorough and careful additions and revisions to this manuscript in response to reviewer feedback. However, one major issue that requires further clarification is the data analysis:

While it's now clear that the authors used their Nanopore App for analysis, the procedure for identifying secondary peaks is still completely elided: "a multi-step routine was used to identify the secondary peak". What are the multiple steps? Are they different for different types of secondary peaks? Please include a description of secondary peak identification analysis in the Methods section; this entire paper is about the subpeaks, so the method for statistically identifying and characterizing them should be explained thoroughly and clearly.

We appreciate the feedback, and as a result, we have expanded upon the explanation of our secondary peak analysis methodology in the Supplementary Information, as well as in the sections dedicated to data analysis and subpeak analysis. These sections now provide a more detailed account of the threshold employed for defining subpeaks in this paper.

Note that in the SI Note "Data Analysis", the analysis step #s do not line up with the #s in the main text (example: main text analysis step 2 is "Track and subtract the baseline" while SI Note step 2 is "Resample and Filter Trace"). Moreover, in the SI Note "Data Analysis" the only mention that is made of sub-peak finding is currently in SI step #6 (the SECOND SI step #6, "Subpeak Analysis") and says just that "By clicking the 'Auto Subpeaks' analysis button, the subpeak details for all events containing subpeaks will be compiled and listed. Thresholds are employed to distinguish genuine positive events from potential false positives caused by folding signals." This does not explain how the analysis is done - what thresholds and statistical tests are used to identify and screen potential subpeaks.

Thank you for pointing this out, we have revised the main text to align the steps with the SI. We included an explanation in SI of what thresholds and statistical tests are used to identify potential subpeaks.

Is the figure on p. 46 of the SI (no caption) intended to be an additional SI figure related to the subpeak finding? If so, it needs a caption and explanation, and a clear reference within the SI note. It appears to show examples of subpeaks and folds rather than explaining their statistical determination. And, given that it is showing a data set in which only 7 events have subpeaks, what is the reader meant to learn from the histograms of those seven events into five bins ("subpeak current")?

We added the caption. We also removed the histograms of the subpeak here.

On a related note, the authors' assertion in the response to reviewers that "The width or height of the subpeaks does not impact the conclusions drawn from the data because we rely on the ratio of bound to unbound events to distinguish positive and negative events" does not make sense since they (presumably; methods unclear) rely on width and height of subpeaks to differentiate folded states from bound protein subpeaks.

We regret any confusion our initial response may have caused. To clarify, following the selection of positive events, we observed slight variations in the subpeak's width and height. This finding aligns with the answer we previously provided concerning translocation speed. However, these variations in

the subpeak parameters, are significantly smaller than and distinct from the DNA folding signal, and in effect do not interfere with the subpeak ratio calculation. Our subpeak selection methodology leverages multiple parameters, including fractional position and either subpeak width or height. These parameters aid in distinguishing potential signals that differ from the DNA folding signal.

We have therefore included a revised supporting figure S17 (now as Supplementary Fig.8 due to the order changes) and caption along with the addition of text in the manuscript (page 5 paragraph 2 for S protein and page 7 paragraph 2 for N protein) to clarify how we dealt with folded events. We also included an additional Supplementary Fig. 17 (now as Supplementary Fig.8), that shows detection and subpeak analysis of S and N protein and folded DNA probe. The fractional peak position histograms are used for the discrimination between folded DNA probe and bound N protein.

The authors also state that they "have made some raw data available for reference" but I do not see this in the review materials- only figure data files of processed data.

Unfortunately for the previous submissions we could not upload the data with the NPG repositories available for review material, due to size limitation (the data are 31.4 gigabytes). Now we have uploaded example raw traces to ZENODO general database (<https://zenodo.org/record/8143395>)

Reviewer #3 (Remarks to the Author):

Regrettably, some of the concerns are not successfully addressed. In general, the authors answered most questions only in the response letter, but failed to incorporate the explanations into the paper to improve scientific rigor.

The authors explained the issues of partial folding of DNA probe sufficiently. Please implement this explanation in the manuscript so the readers may understand why additional optimization to eliminate folding is not necessary as it does not impact result in interpretability.

Based on the previous revisions we added a discussion on the issues of partial folding on Page 5. However, we agree it was not as clearly presented as we would have liked. We have therefore included a revised Supplementary Figure S8 and caption along with the addition of text in the manuscript (page 5 paragraph 2 for S protein and page 7 paragraph 2 for N protein) to clarify how we dealt with folded events.

In the multiplex gene detection section, the current text implies simultaneous quantification which is not achieved. Please clarify, in the main text, that the calibration curve was constructed for N gene, and that the other two binding sites were used to verify the presence of the other 2 genes. This is only vaguely mentioned in the figure caption in the current manuscript.

We have added further clarification on this subject in the main text to avoid confusion (highlighted in Page 9, Paragraph 2).

Please add the rationale of the need of PCR to the manuscript. This is only explained in the response letter but not incorporated into the paper.

We have amended the explanation in the main text (Page 8, Paragraph 2).

Please acknowledge the difference between the buffers/matrices used for calibration curve and the other experiments in the manuscript. The lack of this information will cause confusion and reproducibility for potential readers.

We added a clarification on this difference to avoid confusion (highlighted in the last paragraph in Page 13).

There is no response to this concern: "It appears that Figure 6av is the same as Figure 2d and 3b; Figure 6bv is the same as Figure 4e. The same data should not be presented twice with different figure number references." The repetitive figures need to be removed.

We have removed the Figure 6a(v) and Figure 6b(v) and referred to Figure 2d/3b or Figure 4e.

The ethics statement is still inadequate. The most important information is the IRB approval. Whether this is a human study or not (only using redundant samples as materials without identifying information) should be determined by an Institutional Review Board or an equivalent committee, and the approved project number should be included in the ethic statement.

We amended this ethic statement as follows: "The viral samples were provided from the Imperial College London testing scheme and were from fully anonymised, redundant samples (i.e., samples left over after testing) and retained for assay development, quality assurance, and validation. The sequencing was part of the Imperial College London's response to ensure that new variants were detected and to detect otherwise unexplained clusters. The consent to providing the sample and to the testing of the sample was provided at test booking through an online process. The use of the virus was in accordance with RCPATH guidelines."

In the UK an IRB would not be relevant for use of these samples. Guidance from the Royal College of Pathologists, which is relevant in our case, state that left-over sera or plasma should be stored for as long as practicable, to provide an array of material for future research and disease surveillance purposes. While long-term storage may be impractical in many settings, virology centres and laboratories involved routinely in public health activities should retain sera for a minimum of one year to facilitate 'look-back' exercises, identification of emerging infections and vaccine programme monitoring. Samples that do not contain human cells are not regulated as human tissue by the Human Tissue Act, although ethical constraints on appropriate storage and use nevertheless apply.

REVIEWER COMMENTS

Reviewer #2 (Remarks to the Author):

My thanks to the authors for their updates. In my view, the description of subpeak finding and identification is still inadequate, and has barely been updated as far as I can tell. I have included some specific suggestions below - again - in case the authors see fit to add this information.

As far as I can tell, only vague additional language was included to describe finding the subpeaks. Your methods section still says only that "a multi-step routine was used to identify the subpeaks". What routine? On what basis are subpeaks identified? WHAT are the multiple steps??? If your next sentence "Parameters such as secondary peak amplitude, dwell time, and fractional position were determined" is related to identifying subpeaks as protein or DNA, you need to be more specific. What ranges of these values, or what thresholds, were used to differentiate protein and folded DNA? In what experiments were additional separating parameters needed, and did the cutoffs vary between experiments?

Similarly, in the SI you still say only "By clicking the 'Auto Subpeaks' button, relevant information..." What does the 'Auto Subpeak' button DO, statistically? You say "thresholds could be employed"... but WERE they employed? If so, what thresholds? Were the thresholds different across data sets? If so, why?

REVIEWER Response

Reviewer #2 (Remarks to the Author):

My thanks to the authors for their updates. In my view, the description of subpeak finding and identification is still inadequate, and has barely been updated as far as I can tell. I have included some specific suggestions below- again- in case the authors see fit to add this information. As far as I can tell, only vague additional language was included to describe finding the subpeaks. Your methods section still says only that "a multi-step routine was used to identify the subpeaks". What routine? On what basis are subpeaks identified? WHAT are the multiple steps??? If your next sentence "Parameters such as secondary peak amplitude, dwell time, and fractional position were determined" is related to identifying subpeaks as protein or DNA, you need to be more specific. What ranges of these values, or what thresholds, were used to differentiate protein and folded DNA? In what experiments were additional separating parameters needed, and did the cut-offs vary between experiments?

Similarly, in the SI you still say only "By clicking the 'Auto Subpeaks' button, relevant information..." What does the 'Auto Subpeak' button DO, statistically? You say "thresholds could be employed"... but WERE they employed? If so, what thresholds? Were the thresholds different across data sets? If so, why?

Thank you for the feedback on our paper. We have taken into account your concerns regarding the description of subpeak finding and identification. Here is a list of the changes that address the points raised:

1. Clarifications made in the previous version: contrary to the perception, we have made significant amendments in our previous revision. For instance:
 - a. The SI – data analysis section now provides a comprehensive account of how the peaks are determined.
 - b. We updated the SI figure S8 to emphasise the subpeak analysis.
 - c. The SI text has undergone significant modifications to incorporate statistics related to the subpeaks.

In this version, we have included an even more detailed discussion that can be found in Supplementary Fig. 36-37

2. Description of 'Multi-Step Routine':
 - a. We acknowledge that the term "multi-step routine" was ambiguous. We meant it to encompass all stages of data analysis. We removed the term and described the analysis step-by-step.
 - b. The subpeak identification relies on several parameters like peak height, dwell time, and fractional position.
 - c. The details and all the parameters for each target experiment are described below and in the Data Analysis section in Supplementary Information (Supplementary Fig. 36-37).
3. Detailing the thresholds for each target:
 - a. We have incorporated descriptive paragraphs and figures that clearly lay out the criteria for subpeak selection. This addresses:
 - i. How each parameter is employed.

- ii. The supporting schematic of the parameters used to identify positive subpeaks.
 - iii. The threshold values for each parameter when choosing various signals.
- b. An in-depth explanation of thresholds applied for specific detections (S protein, N protein) has been included, explaining the rationale behind the chosen thresholds.

In the updated SI with additional discussion and supplementary Fig. 36, we illustrate what parameters and thresholds are used and how they are defined:

The last step of the data analysis involves extracting the subpeak information. By clicking the 'Auto Subpeaks' button, relevant information such as subpeak amplitude, subpeak dwell time, the fractional position of the subpeak, fractional width of the subpeak, number of subpeaks, location of subpeaks could be extracted directly from the CUSUM fits which has been illustrated in Step 5. Due to DNA folding, thresholds could be employed to distinguish between genuine positive events and potential false positives. The fractional position is often used to isolate events associated with subpeaks originating at a defined location. As the subpeak width for the protein is typically smaller than that of folded DNA, this can also be used as a threshold. Peak amplitude can also be used as a discriminator (**Supplementary Fig. 36**).

Supplementary Fig. 36 | The parameters used for identifying positive subpeak.

a. The schematic of the key parameter for select positive events in this experiment. b. The threshold value for each parameter in the selection of different signals.

Then, we illustrate the detailed thresholds for each target and the reason for choosing them:

In detail, for S protein detection, the threshold for the normalised subpeak position (The 'Fractional Peak' in the Nanopore App interface) has been set to 0.1 ± 0.1 and 0.9 ± 0.1 as the S protein would bound to the SBA located at the end of the molecular carrier. To discriminate the S protein signal and partially folded signal, which is observed at the events' beginning or end, another two thresholds, subpeak width and subpeak/DNA ratio, have been applied. For partial DNA folding, the subpeak always shows a wide rectangular shape with a longer dwell time (0.447 ± 0.231 ms) and is very unlikely to show a narrow, sharp spike due to the limitation with the persistence length (> 50 nm). However, the S protein binding results in a short dwell time and high current amplitude secondary peak due to the relatively larger size of the S1 protein compared to the folded DNA. Here, the threshold for subpeak width was set to <0.2 ms to select the events with narrow subpeaks where the threshold for subpeak/DNA ratio was set to >1.5 as the folded DNA subpeak/unfolded DNA level was 0.662 ± 0.027 (**Supplementary Fig. 36**). It should be noted that when the translocation event exhibited both folding and protein binding signals, these events were individually examined to verify that all binding events were classified correctly.

The representative DNA folded events that can be filtered out, the S protein bound events that can be selected automatically, and the S protein bound events that contain DNA folding, which requires a manual check, were listed in **Supplementary Fig. 37**.

The threshold for the normalised subpeak position for N protein detection has been set to 0.5 ± 0.2 . This is in good agreement with the placement of N protein binding site, which is designed to have a fraction position of 0.48 based on the sequence on the molecular carrier. Subpeak amplitude was not used to discriminate between events due to the similarity in amplitude between folded and protein-bound events. The observation of DNA knots in the middle of the translocation event is uncommon ($< 0.1\%$), and hence, all protein-bound events could be isolated based on normalised peak position and subpeak width (< 0.3 ms). Similarly, all folded events were cross-checked manually to ensure all binding events were counted as partial folding will lead to a slight shift in the fractional position. For RNA 1.0 site, we use a similar strategy as this is N amplicon site, and it is responsible for confirmation of the positive signal and RNA quantification. Therefore, in addition to applying a 0.1 ± 0.1 and 0.9 ± 0.1 normalised subpeak threshold and < 0.2 ms subpeak width threshold, the events were always checked manually to ensure accurate counting.

4. It should be noted that across different nanopores, the threshold values remain the same; the reason is:
 - a. normalised subpeak position would not be affected significantly by the nanopores with different sizes/shapes;
 - b. the dwell time difference in nanopores with a size in a very narrow range is very subtle;
 - c. The change in peak amplitude values caused by different nanopores is most evident. However, we used the ratio of Subpeak to DNA level as a threshold and utilised normalisation to eliminate this effect. In **Supplementary Fig. 37**, we have shown 32 events collected using different nanopores and classified using the same thresholds.

5. We also added one figure showing that using the thresholds, the DNA false positive signal can be filtered out, the protein-bound real positive signal can be selected automatically, and the folded DNA-bound protein signal needs to be checked manually. We also illustrated that the cut-off values did not vary in different devices and explained the reason above.

Supplementary Fig. 37 | Example events that have been isolated using the thresholds.

Utilising specific thresholds, we can distinguish and separate events with relevant subpeaks. Signals from molecular carriers, regardless of their folded or unfolded state, are excluded (indicated in the red box). Events that satisfy all threshold criteria for subpeak information are identified as potential positive signals (highlighted in the green box). For events exhibiting both folding and protein-binding indications, individual assessments were made to ensure proper classification of all binding instances (shown in the yellow box). The events we listed here are all recorded with different nanopores.

Clicking the 'Auto Subpeaks' button will classify individual events based on the information obtained from the CUSUM fit. A CUSUM fit (represented by the red dashed line) is employed for each event. Different molecules, such as DNA protein, cause different disruptions in the current. The resulting signal has "steps" corresponding to these different molecules' size and charge. The CUSUM algorithm is used to detect these "steps". Once these significant changes or steps are detected, the continuous signal can be segmented into "events", each event corresponding to a particular DNA, protein or DNA/protein-bound complex.

In summary, we have made deliberate efforts to provide a more in-depth explanation of our methodology, supported by revised supplementary figures. We hope these clarifications and additions address the reviewer's concerns, and we appreciate the opportunity to enhance the clarity and thoroughness of our work.

REVIEWERS' COMMENTS

Reviewer #2 (Remarks to the Author):

My thanks to the authors for providing the details of subpeak analysis. I will be delighted to see this publication in Nature Communications in the near future.